**Implications of different nitrogen input sources for potential production and carbon flux estimates in the coastal Gulf of Mexico (GOM) and Korean coastal waters**

Jongsun Kim [a*] Piers Chapman [a, c], Gilbert Rowe [a, b], Steven F DiMarco [a, c], and Daniel C. O. Thornton[a]

[a] Department of Oceanography, Texas A&M University, College Station, TX 77843-3146, USA

[b] Department of Marine Biology, Texas A&M University, Galveston, TX 77553, USA

[c] Geochemical and Environmental Research Group, Texas A&M University, College Station, TX 77843-3149, USA

[*] Corresponding author

*J. Kim. Email: jongsun@tamu.edu

Submit to Ocean Sciences

**Abstract**

The coastal Gulf of Mexico (GOM) and coastal sea off Korea (CSK) both suffer from human-induced eutrophication.   We used a nitrogen (N) mass balance model in two different regions with different nitrogen input sources to estimate organic carbon fluxes and predict future carbon fluxes under different model scenarios.   The coastal GOM receives nitrogen predominantly from the Mississippi and Atchafalaya Rivers and atmospheric nitrogen deposition is only a minor component in this region.   In the CSK, groundwater and atmospheric nitrogen deposition are more important controlling factors.   Our model includes the fluxes of nitrogen to the ocean from the atmosphere, groundwater, and rivers, based on observational and literature data, and identifies three zones (brown, green and blue waters) in the coastal GOM and CSK with different productivity and carbon fluxes.   Based on our model results, the potential primary production rate in the inner (brown water) zone are over 2 gC $m^{-2}$ $day^{-1}$ (GOM) and 1.5 gC $m^{-2}$ $day^{-1}$ (CSK).   In the middle (green water) zone, potential production is between 0.1 to 2 (GOM) and 0.3 to 1.5 gC $m^{-2}$ $day^{-1}$ (CSK).   In the offshore (blue water) zone, productivity is less than 0.1 (GOM) and 0.3 (CSK) gC $m^{-2}$ $day^{-1}$.   Through our model scenario results, overall oxygen demand in the GOM would increase approximately 21% if we fail to reduce riverine N input, likely increasing considerably the area affected by hypoxia.   Comparing the results from the U.S. with those from Korea shows the importance of considering both riverine and atmospheric inputs of nitrogen.   This has direct implications for investigating how changes in energy technologies can lead to changes in the production of various atmospheric contaminants that affect air quality, climate and the health of local populations.

**Keywords:**

Chemical tracers, Biological processes, Shelf-seas, Gulf of Mexico, Yellow Sea.

## Introduction

Industrial expansion and anthropogenic emissions are major factors leading to increased
coastal productivity and potential eutrophication (Sigman and Hain 2012). Coastal primary
production is controlled largely by nitrogen (N) and phosphorus (P), and the relative supply of
each determines which element limits production (Paerl 2009); freshwater inputs and the
distance from sources such as river mouths are also important (Dodds and Smith 2016).
Changes in nutrient loading from air-borne, river-borne and groundwater sources can also affect
which element limits coastal productivity (Sigman and Hain 2012). Most coastal regions are N-
limited, however, at certain times conditions can change from N-limited to P-limited (Dodds and
Smith 2016; Howarth and Marino 2006). Sylvan et al. (2006), for example, suggested that the
coastal GOM, especially near the Mississippi River delta mouth, is P-limited at certain times.
Several studies have shown that increasing atmospheric nitrogen deposition (AN-D) is
contributing to ocean production globally, including to eutrophication, and is potentially of
future importance in the GOM (Cornell et al., 1995; Doney et al., 2007; Duce et al., 2008; He et
al., 2010; Kanakidou et al., 2016; Kim 2018; Kim (TW) et al., 2011; Lawrence et al., 2000; Paerl
et al., 2002). Recently, Kim (TW) et al. (2011), using a model simulation showed that AN-D
controls approximately 52% of the coastal productivity in the Yellow Sea. Global NOx
emissions have increased but appear to be changing differently in the US and Asia (Kim (JY) et
al., 2010; Luo et al., 2014; Shou et al., 2018; Zhao et al., 2015), and may affect not only coastal
productivity but also global total nitrogen budgets. This study uses a box model to define
potential carbon fluxes based on different nitrogen input sources in two different regions, the
Coastal Gulf of Mexico (GOM) and the Coastal Sea off Korea (CSK). The GOM and CSK
were selected in this study because while the major input source to the coastal ocean in both
regions is riverine, the AN-D and submarine groundwater discharge (SGD) are considerably
more important in the CSK region (Wade and Sweet, 2008; Zhao et al., 2015).

Most previous model studies in the GOM have been used to predict the size of the

hypoxic zone (e.g., Fennel et al., 2006, 2011, 2013; Green et al., 2008; Hetland and DiMarco
2008; Justic et al., 2002; Scavia et al., 2004; Turner et al. 2006, 2008), although Bierman et al.
(1994), used a mass balance model to estimate carbon flux and oxygen exchange.    The mass
balance model is a useful tool to calculate nutrient or carbon fluxes and to estimate production in
the coastal ocean (Kim (JS) et al, 2010; Kim (G) et al., 2011), and such models have been
successfully used in many regions and individual coastal systems to estimate ecosystem
metabolism, e.g., in the Patuxent River estuary of the Chesapeake Bay (Hagy et al. 2000; Testa
et al., 2008) and in the LOICZ (Land Ocean Interactions in the Coastal Zone) project (e.g.,
Ramesh et al., 2015).    However, there are few such model studies in the GOM and CSK.    All
previous models for the GOM and the CSK have considered only riverine N as the predominant
input source, and no one has considered AN-D as an input in either region.

In this study, we aimed to: 1) build a mass balance model considering not only riverine N

input but also air-borne and groundwater-borne N; 2) use it to calculate potential primary
production in the three regions defined by Rowe and Chapman (2002, henceforth RC02, see next
section) and their associated coastal productivity; and 3) use the mass balance model to test the
RC02 hypothesis.    Because RC02 did not quantify their model with nutrient data and no one
has applied this model to another region, we tested the RC02 hypothesis using data from both the
GOM and the CSK that include low salinity samples.    We used historical data from the mid-
western part of the CSK and evaluated the theoretical model of RC02 in both areas where
freshwater with high terrestrial nutrient input mixes into the coastal ocean.

**Study areas**

The Texas-Louisiana (LATEX) shelf in the northern Gulf of Mexico is affected by
coastal nutrient loading, leading to hypoxia, coming from two major terrestrial sources (the
Mississippi and Atchafalaya Rivers that together form the Mississippi-Atchafalaya River System
MARS).   These two major rivers have different nutrient concentrations.   The Gulf of Mexico
(GOM) is a semi-enclosed oligotrophic sea and the MARS is the major source of nutrients and
freshwater to the northern GOM (Alexander et al., 2008; Rabalais et al., 2002; Robertson and
Saad, 2014).   The MARS drains 41% of the contiguous United States (Milliman and Meade,
1983) and discharges approximately 20,000 $m^3$ $s^{-1}$, or about 60% of the total freshwater flow,
(about 10.6 x $10^{11}$ $m^3$ $year^{-1}$ or 3.4 x $10^4$ $m^3$ $s^{-1}$) to the northern side of the GOM.   The
remainder comes from other U.S. rivers, Mexico and Cuba (Nipper et al., 2004).
At the Old River Control Structure on the lower Mississippi River approximately 25% of
the Mississippi River's water is diverted into the Atchafalaya River, where it mixes with the
water in the Red River.   The flow in the Atchafalaya River totals 30% of the total MARS flow
(Figure 1a).   Several projects have investigated the relationship between nutrients and the
marine ecosystem, and how this leads to hypoxia in the GOM (e.g. Bianchi et al., 2010; Diaz and
Rosenberg, 1995, 2008; Forrest et al., 2011; Hetland and DiMarco, 2008; Laurent et al., 2012;
Quigg et al., 2011; Rabalais and Smith, 1995; Rabalais et al., 2007; Rabalais and Turner 2001;
Rowe and Chapman 2002).   Strong stratification due to the high freshwater discharge from the
MARS, local topography (DiMarco et al., 2010), wind direction, and high nitrate concentration
all affect hypoxia formation, with upwelling-favorable wind facilitating its development (Feng et
al., 2012, 2014).
In the Northern GOM, the major factor controlling coastal productivity is riverine N input.
Rowe and Chapman (2002), defined three theoretical zones over the LATEX shelf close to the
Mississippi and Atchafalaya River mouths to predict the effects of nutrient loading on hypoxia
along the river plumes and over the shelf.    They named these the brown, green, and blue zones
(Figure 2).    Nearest the river mouths is a 'brown' zone, where the nutrient concentrations are
high, but the discharge of sediment from the river reduces light penetration and limits primary
productivity within the plume.    Further away from the river plume is a stratified 'green' zone
with available light and nutrients that result in high productivity.    In this region, the rapid
depletion of nutrients is due to biological uptake processes that depend on the season and river
flow (Bode and Dortch, 1996; Dortch and Whitledge, 1992; Lohrenz et al., 1999; Turner and
Rabalais, 1994).    Still further offshore, and also along the river plume to the west, there is the
so-called 'blue' zone, defined arbitrarily by nitrate concentrations of 1 µM or less, which is
dominated by intense seasonal stratification and a strong pycnocline, so that in the surface layer
nutrients are limiting at this distance from the rivers and most primary production is fueled by
recycled nutrients (Dortch and Whitledge, 1992).    It is important to note that RC02 makes clear
that the edges of the zones (geographical regimes) are not static, but change over time depending
on season, river flow, and biological processes (Figure 2).
The coastal sea off western Korea (CSK) forms the eastern side of another semi-enclosed
basin (the Yellow Sea) and is affected by freshwater discharge from river plumes in the same
way as the coastal GOM, although the freshwater flow is considerably less.    The Yellow Sea
covers about 380,000 km$^2$ area with an average water depth of 44 m, and numerous islands are
located on its eastern side (Liu et al., 2003).    Our specific study area is the mid-western coastal
region from the Taean Peninsula to Gomso Bay (Figures 1c and 1d).
There is a strong tidal front in the coastal area near the Taean Peninsula due to sea floor
topography and the coastal configuration (Park, 2017; Park et al., 2017).   The region also
contains several bays (Garolim Bay, Gomso Bay and Cheonsu Bay), and is affected by
discharges from a large artificial lake (Saemangeum lake) as well as the freshwater discharge
from the Keum river plume that contains high concentrations of nutrients (Lim et al., 2008).
Conditions in the mid-western CSK near the Taean peninsula are similar to the coastal GOM,
because of mixing of two different water masses from Gyunggi Bay (Han River) and the Keum
River (Choi et al., 1998, 1999).   The annual mean flow rates within the Keum River were about
70 $m^3$ $s^{-1}$ (normal period) and 170 $m^3$ $s^{-1}$ (flood period) (Yang and Ahn 2008). Precipitation
within the catchment was 1,208 mm $year^{-1}$ during 2003 to 2005 (Yang and Ahn 2008).
Unlike the coastal GOM, the CSK has increased nitrogen inputs from atmospheric
nitrogen deposition (AN-D, which is approximately five times higher than in the GOM, Table 2)
(Kim (JY) et al., 2010; Luo et al., 2014; Shou et al., 2018; Zhao et al., 2015) and nutrient inputs
from the groundwater discharge (Kim (JS) et al., 2010; Kim (G) et al., 2011).   AN-D has
increased in the CSK owing to industrial development in China during the last few decades,
which has led to increased atmospheric N emission.

**Data and Methods**
*Riverine N data*
Hydrographic data from the MCH (Mechanisms Controlling Hypoxia – MCH Atlas)
projects in the Gulf of Mexico were collected from the National Oceanographic Data Center
(https://www.nodc.noaa.gov) covering the period from 2004 through 2007 (Table 1).   We
excluded cruises MCH M6 and M7 because the threat of hurricanes led to sampling stations in
different areas from the other cruises.    The study sites and sampling areas are shown in Figure
1b.    Quality control removed inconsistencies and anomalies in the data (e.g., removing outliers,
missing data found by linear interpolation).    Hydrographic data from the CSK (nutrients,
salinity, oxygen) were collected during several cruises (Table 1 and Figure 1c and 1d), and the
data were put through similar QA/QC routines.

*Atmospheric Nitrogen Deposition (AN-D) data*

AN-D data from around the US are sparse (Table 2).    Most US data have been collected

along the east coast of the US and the only data in the GOM region were collected near Corpus
Christi (~1 g m$^{-2}$ year$^{-1}$; Wade and Sweet, 2008), Considerable AN-D could be expected,
however, from the large number of petrochemical and fertilizer plants in southern TX, especially
near Houston and along the Mississippi.    While there are more data from the Yellow Sea (Kim
(JY) et al., 2010; Luo et al., 2014; Shou et al., 2018; Zhao et al., 2015), they are still limited
owing to the broad sampling coverage.    While AN-D data in the Asian region were up to 14 g
m$^{-2}$ year$^{-1}$, data from the eastern side of the US were under 1 g m$^{-2}$ year$^{-1}$, even lower than in the
GOM, suggesting there is currently not a large contribution from AN-D to total N loads to the
North Atlantic Ocean.    The approximate order of magnitude difference in AN-D concentrations
between the GOM and the CSK is due to the continuing industrial development in East Asia and
the resulting N emissions (Wang et al., 2016; Zhao et al., 2015).    Lamarque et al., (2013)
reported model results, which covers our study regions, and their model appears to underestimate
AN-D at the sampling sites compared with observational data in the GOM (Wade and Sweet,
2008).    However, the pattern of AN-D inputs between GOM and CSK from Lamarque et al.,
(2013) shows around five times difference between the two regions, which agrees with our data.
Thus, in our model, we used observational data for both regions, as shown in Table 2.

*Methodology: N-mass balance model*

Our model consists of three sub-regions based on sampling locations during MCH cruises

(Figure 3), each of which contains a series of one-quarter degree square boxes, as followed by
Belabbassi (2006).    The quarter degree boxes in this study were separated into an upper box and
a lower box, based on pycnocline depth, as defined by a sharp change in density and coincides
generally with a minimum change in oxygen concentration of 22.33 µM.    We assume steady
state conditions, and estimate potential production, which we count as an estimate of potential
carbon flux (Figure 3a).    Primary production (PP) above the pycnocline is expected to be higher
than below it (Anderson 1969; Sigman and Hain, 2012), which means that the two layers have
different production rates.    The difference in PP between upper and lower boxes also depends
on the freshwater discharge rate, which determines nutrient input to the upper layer, seasonal
variability, and transfer processes between the layers.    While chlorophyll can be found below
the pycnocline (DiMarco and Zimmerle, 2017), the fact that it is typically associated with low
oxygen concentrations suggests that the phytoplankton are either inactive or, more likely,
producing at a very slow rate.

The N-mass balance box model is modified from previous models to calculate the net

removal of dissolved inorganic nitrogen (DIN) inside each box, which represents potential
primary production (PPP) (De Boer A.M. et al., 2010; Kim (G) et al., 2011) (Equation 1).    In
this model, DIN concentration includes ammonium ($NH_4^+$), nitrate ($NO_3^-$), and nitrite ($NO_2^-$).

$$F_{River}^{DIN} + F_{Atmo}^{DIN} + F_{Bott}^{DIN} - F_{Export}^{DIN} - F_{Deni}^{DIN} = F_{Removal}^{DIN} - \text{Eq. 1}$$

163 where, $F_{River}^{DIN}$, an input term, is DIN flux from each river discharge and calculated with $C_{Box}^{DIN}$,

164 the DIN concentration in each box, $A_{Bott}$, the bottom area of each quarter degree box, and

165 $F_{River}$, river discharge rate $(C_{Box}^{DIN} \times A_{Bott} \times F_{River})$. As another input term, $F_{Atmo}^{DIN}$ is the

166 flux from atmospheric nitrogen deposition. $F_{Bott}^{DIN}$, the benthic flux is additional input term in

167 the sub-pycnocline layer box. The one quarter degree blue boxes located closest to the

168 Mississippi and Atchafalaya river mouths were assumed to be the only ones affected by riverine

169 input (Figure 3b). As an output term, $F_{Export}^{DIN}$ as an advection term was calculated from the

170 current velocity in each region from observations (Nowlin et al., 1998a, b) and from literature

171 data (Jacob et al., 2000; Lim et al., 2008) and the exchange between boxes from the residence

172 time in each box. Note that water and nutrient exchange can take place through all four sides of

173 each box, so the array is two-dimensional. $F_{Export}^{DIN}$ for water mixing was calculated from these

174 factors; $C_{EX}^{DIN}$ is the difference in DIN concentration between adjacent boxes, $V_S$ is the water

175 volume of each box, and $\lambda_{Mix}$ is the mixing rate of each box $(C_{EX}^{DIN} \times V_S \times \lambda_{Mix})$. We used

176 a reciprocal of the water residence time that we considered to represent horizontal mixing, i.e.

177 dispersion. Another output term is $F_{Deni}^{DIN}$, denitrification process from the water column, and

178 $F_{Removal}^{DIN}$ is removal by biological production. The details of the model definitions are given

179 below in Table 3 and shown in Figure 3. Each arrow indicates input (blue) and output (red)

180 terms (Figure 3). Input/output terms vary based on whether the boxes are above/below the

181 pycnocline, while there are separate inputs from the Mississippi and Atchafalaya rivers in the

182 GOM and Keum and Han rivers in the CSK, respectively.

In order to calculate the net removal of DIN in a box above the pycnocline layer, we used
our N-mass balance model in Equation 2.

$$F_{River}^{DIN} + F_{Atmo}^{DIN} - F_{Export}^{DIN} - F_{Sink}^{DIN} = F_{Removal}^{DIN} - \text{Eq. 2}$$

The boxes above the pycnocline layer have two input terms:    1) $F_{River}^{DIN}$, riverine N,
which affects only a subset of boxes along the edge of each region, and 2) $F_{Atmo}^{DIN}$, atmospheric
nitrogen deposition (AN-D), which affects every box equally.    The mean value of Asian data, as
shown in Table 2 (Kim (JY) et al., 2010; Luo et al., 2014; Shou et al., 2018; Zhao et al., 2015), is
used for $F_{Atmo}^{DIN}$ of the CSK region, which is initially five times higher than that of the GOM (1.4
X $10^5$ mol day$^{-1}$; Wade and Sweet, 2008).    We also considered vertical sinking as an input for
the sub-pycnocline layer box and as an output from the upper layer.    Other possible input
factors might be upwelling/downwelling processes; however, these factors are neglected in the
model because both regions are shallow and close inshore (Feng et al., 2014; Lim et al., 2008)
and we have no observational data on upwelling/downwelling rates.    The output terms are the
following: 1) $F_{Export}^{DIN}$, the exchange rate between each box (obtained from the different N
concentrations in each box and the mass transfer between them), and 2) $F_{Sink}^{DIN}$, removal by
biological production, including sinking (assuming that any other removal factors are neglected
above the pycnocline).    We tested the RC02 three zone hypothesis in the upper box layer, in
which we can also examine the horizontal influence (horizontal extent) of the river plume based
on production rates.
Below the pycnocline layer we used the revised Equation 3.

$$F_{Bott}^{DIN} + F_{Sink}^{DIN} - F_{Export}^{DIN} - F_{Deni}^{DIN} = F_{Removal}^{DIN} \quad - \text{Eq. 3}$$

Equation 3 has two separate input terms; 1) The benthic flux $F_{Bott}^{DIN}$ term contains all the

potential input from the bottom sediment (defined here as net DIN release from the bottom
sediment) including nutrient regeneration by bacteria, groundwater nutrient inputs, and an uptake
of nitrate ($NO_3^-$) and nitrite ($NO_2^-$) mainly by sedimentary denitrification (McCarthy et al., 2015;
Nunnally et al., 2014), and 2) $F_{Sink}^{DIN}$ term as a vertical sinking from the box above the
pycnocline layer, for which we used data from Qureshi (1995). The unit of $F_{Sink}^{DIN}$ was
converted to mol day$^{-1}$ from the unit of original data (gN m$^{-2}$ day$^{-1}$) with area of box (0.25 m x
0.25 m) and molar mass of N (14 g mol$^{-1}$).

In the GOM, benthic sediments provide excess ammonium to overlying water by

regeneration processes such as remineralization (Lehrter et al., 2012; Nunnally et al., 2014;
Rowe et al., 2002). Generally, there is an uptake of nitrate and nitrite mainly by sedimentary
denitrification (McCarthy et al., 2015) or dissimilatory nitrate reduction to ammonium (DNRA)
and assimilation by benthic microalgae (Christensen et al., 2000; Dalsgaard, 2003; Thornton et
al., 2007). Due to this, net DIN flux was used as the value of $F_{Bott}^{DIN}$, which shows DIN release
from bottom sediments to overlying water column. For example, in the GOM, the sum of
nitrate and nitrite fluxes to bottom sediments (e.g., May: -10.05, July -61.9, August: -48.42 μmol
N m$^{-2}$ h$^{-1}$) were similar or smaller than the flux of ammonium from bottom sediments (e.g., May:
203, July: 152, August: 156 μmol N m$^{-2}$ h$^{-1}$) off Terrebonne bay (McCarthy et al., 2015). In
the CSK, the sum of nitrate and nitrite flux to bottom sediments and ammonium flux are 0.5 ~
1.4 mmol N m$^{-2}$ d$^{-1}$ and 1.3 ~ 9.6 mmol N m$^{-2}$ d$^{-1}$, respectively, which indicated that excess
ammonium with additional nitrate and nitrite were released from sediments in this region (Lee et
al., 2012). The release of nitrate and nitrite in the CSK unlike the GOM can be estimated due to
high inputs of nitrogen by groundwater in the CSK (Kim (G) et al., 2011) even though there is
minor uptake of nitrate and nitrite. Diffusion from groundwater can probably be ignored in the
GOM as Rabalais et al. (2002) reported that the groundwater discharge is very low in coastal
Louisiana, but is likely important elsewhere and is known to be important in the CSK. Based
on this, we averaged and sum the fluxes data of nitrate, nitrite, and ammonium from McCarthy et
al., 2015 for the GOM and Lee et al., 2012 for the CSK, respectively, and then applied
$F_{Bott}^{DIN}$ value as 1.2 mmol N m$^{-2}$ day$^{-1}$ in the GOM and 6.2 mmol N m$^{-2}$ day$^{-1}$ in the CSK. Thus,
in equation 3, the benthic flux term is calculated from existing literature results after considering
all DIN fluxes as above (Lee et al., 2012; McCarthy et al., 2015), and then multiplied by the area
of each box.
The output terms are; 1) $F_{Export}^{DIN}$, the exchange rate between each box in the lower layer,
and 2) $F_{Deni}^{DIN}$, the denitrification rate from the water column. Due to high stratification at the
pycnocline, upward transfer of dissolved material from the lower layer to the upper layer is
assumed not to occur in our model. Also, denitrification from the water column below the
pycnocline is a significant N removal process, which removes up to a maximum 68% of total N
input from the Mississippi River (MR) in the GOM (McCarthy et al., 2015). As the value of
$F_{Deni}^{DIN}$ in the GOM, we used a direct measurement of denitrification rates from the McCarthy et
al., (2015) in the water column (88 µmol m$^{-2}$ h$^{-1}$, which converted to 2.1 mmol N m$^{-2}$ day$^{-1}$)
where the stations were exactly same as our sub-region A, B, and C. We assumed this applied
only below the pycnocline where oxygen concentrations decrease. However, in the CSK, there
is no water column denitrification data because the dissolved oxygen concentration has never
been down below about 4 mg L$^{-1}$ during our data periods. Based on this, we estimated that
there is a very little water column denitrification in the CSK, so we did not count this term in the
CSK.    Thus, we only considered the sedimentary denitrification term for the CSK region.

Water transport in the region is generally from the east, i.e., from near the Mississippi

River in Sub-region A to the west, near the Atchafalaya River in Sub-region C during non-
summer periods.    During summer, the winds change direction from easterly to westerly,
blocking the water flow to the west (Cho et al., 1998).    We calculated advection from current
meter data collected during the LATEX program (Nowlin et al., 1998a, b) from April 1992 to
December 1994, from which we determined U (west to east flow) and V (south to north flow)
components (cm s$^{-1}$).    Figure 4 shows the mean values of coastal ocean current velocities.    The
annual range of the currents is 0 to 30 cm s$^{-1}$ for the longshore component, with standard
deviation of about 8 cm s$^{-1}$, and 0 to 7 cm s$^{-1}$ for the cross-shelf component, with a similar
standard deviation, but these current velocities are not constant and change depending on time
and day.    The annual current velocities in the CSK are more affected by tidal exchange and the
presence of the Yellow Sea Current, but velocities are similar to those in the GOM (Jacob et al.,
2000; Lim et al., 2008).    The annual range of the currents is around 0 to 28 cm s$^{-1}$ and 0 to 7 cm
s$^{-1}$ for the cross-shelf component.    Thus, we used the mean value of the current velocity for the
time of year during each cruise in both the GOM and the CSK for calculating the advective flow
in both alongshore and onshore/offshore directions.

To run the box model, we assumed three factors: 1) the study area is in a steady state

condition, with equal input sources and outputs, 2) AN-D is evenly distributed across each area,
and 3) DIN is fully utilized by phytoplankton growth in the layer above the pycnocline, so we
can neglect other removal factors.    However, in the layer below the pycnocline, as we
mentioned above, denitrification, which leads to a main loss of DIN as nitrogen gas, is
considered as another output term in Equation 3.   Because we assumed that all DIN removed is
fully consumed by primary production above the pynocline, we can calculate potential carbon
fluxes and oxygen consumption using the Redfield ratio (C: N: $-O_2$: P = 106: 16: 138: 1).   The
PPP can be compared with $^{14}$C measurement data (Lohrenz et al., 1998, 1999; Redalje et al.,
1994; Quigg et al., 2011) and dissolved oxygen data from MCH mooring C at 29° N, 92° W (4
March 2005 ~ 10 July 2005) (Bianchi et al., 2010).

**Results**
*An N-mass balance model for the Texas-Louisiana Shelf*
The existence of the three zones suggested by RC02 has been verified from winter data
using nutrient/salinity relationships (Kim 2018).   Figure 5 shows the contour graph based on the
mean concentration of DIN at each station during the MCH M4 (March 2005) cruise.   For
operational and modeling purposes, stations were grouped into three sub-regions – near the
Mississippi (A), near the Atchafalaya (C) and an intermediate region (B) between ~90°-91°W.
During summer, it is hard to use nutrient/salinity relationships directly because riverine nutrient
inputs are lower and phytoplankton growth causes rapid nutrient consumption over the shelf,
leading to low overall nutrient surface concentrations.   We calculated the mean [DIN] in each
box, and then used the relationship between DIN and salinity to define the edges of the three
zones.   Near the coast salinity was consistently low, with high turbidity from the river water
discharge.   This was labelled the brown (river) zone.
A range of N input values from various sources were used in the N-mass balance model
to estimate PPP and carbon fluxes in the coastal GOM.   The PPP rates were highest near the
river mouth and we set the boundaries of production for each zone based on our N-mass balance
model results and mean [DIN] data.    We defined the brown zone as having the PPP rate of over
2 gC m$^{-2}$ day$^{-1}$ because of the high input of N from the river, AN-D, and benthic fluxes, and the
rate in the blue zone is less than 0.1 gC m$^{-2}$ day$^{-1}$.    The PPP rate in the green zone is then
between 0.1 and 2 gC m$^{-2}$ day$^{-1}$.    Basically, these PPP ranges were set based on synthesized
measured ranges of coastal GOM primary production, as defined for near, mid, and far fields of
the coastal GOM (Dagg and Breed 2003; Lohrenz et al., 1999).    Note that our model results of
the PPP might overestimate the actual production because of light limitation, following RC02.

The edges of the three zones above and below the pycnocline layer, based on our N-mass

balance model results, are shown in Figures 6a and b.    The patterns of the boundaries above and
below the pycnocline differ from the edges of the zones.    The brown zone was found above the
pycnocline on all cruises close to the Mississippi River mouth because of the high nutrient
concentrations, but only appeared off the Atchafalaya River in March 2005 (MCH M4).
However, below the pycnocline it was found only in April 2004 (MCH M1) in sub-region A.
This suggests that vertical transport across the pycnocline rapidly removes the high levels of
suspended material that cause light limitation above the pycnocline.    In the green zones, the
nutrient source is mostly supported directly by the river, with minor additional sources of N from
vertical sinking, AN-D, and benthic fluxes.    We utilized the vertical sinking flux from the
sediment trap data from Qureshi (1995) below the pycnocline layer to estimate PPP.    This
varied between 0.1-1.0 gN m$^{-2}$ day$^{-1}$ (Table 3).    Typically, in the blue zone where biological
production is low, vertical sinking followed by local decomposition is assumed to be the major
factor that changes the nutrient concentration in the lower layer.    The blue zone is always more
extensive below the pycnocline than above it, which suggests there is little or no sub-pycnocline
production except close to the coast and/or the river mouths, and reinforces the assumption that
any chlorophyll below the pycnocline is inactive (Figure 6b).   Thus, we can identify the
horizontal influence of the river plume in the layer below the pycnocline and the variation in the
boundaries of the three zones, based on the observed nutrient data from a bottom layer and our
N-mass balance model.   The model suggests that regions of moderate potential productivity
extend offshore at least as far as 28° 30'N in sub-region B, both above and below the pycnocline.

*An N-mass balance model calibration*

The model calibration was done with historic literature data.   Literature data suggest that

observed PP rates in the green and brown zones of the coastal GOM vary between 0.4 gC m$^{-2}$
day$^{-1}$ (winter) and $\sim$ 8 gC m$^{-2}$ day$^{-1}$ (summer) (Dagg et al., 2007; Lohrenz et al., 1998, 1999;
Redalje et al., 1994).   Recently, Quigg et al. (2011) determined the integrated PP rates with $^{14}$C
measurements during 2004 in the coastal GOM.   The highest integrated PP rates were found
near the Mississippi River at 3.5 gC m$^{-2}$ day$^{-1}$ (in July), and near the Atchafalaya River at 2.7 $\sim$
5.9 gC m$^{-2}$ day$^{-1}$ (in May to July) (in the brown and green zones).   However, lowest integrated
PP rates were on the outer part of the LATEX shelf (the blue zone) at 0.07 gC m$^{-2}$ day$^{-1}$ (in
March), 0.04 $\sim$ 0.15 gC m$^{-2}$ day$^{-1}$ (in May), and 0.33 $\sim$ 0.91 gC m$^{-2}$ day$^{-1}$ (in July).   Additionally,
Quigg et al., (2011) pointed out that these higher PP values were affected by high riverine
nutrients input from the MR that flows westward during that time period.

The actual PP ranges were similar with our model-based PPP (Figure 6).   However, this

was different from RC02's brown zone.   This might be due to the differences between methods
such as $^{14}$C, our N-mass balance model, and RC02's theoretical model.   Typically, RC02
assumed that the brown zone is light limited due to high sediment turbidity, but our model does
not account for this and only considered DIN concentrations.   Except for this, our PPP results
are similar to direct productivity measurements from the $^{14}$C incubations (Quigg et al., 2011).
Our model result (PPP) showed the same range of values as $^{14}$C incubations (e.g., Dagg et al.,
2007; Lohrenz et al., 1998, 1999; Quigg et al., 2011; Redalje et al., 1994) in the three sub-
regions.

Note that our model assumed all the biological uptake could be converted directly to

production rates, which we considered as PPP.   The PPP from cruises MCH M1 ~ M8 for
samples from above the pycnocline calculated using our model is reasonable based on
comparison with previous PP values (Figure 6a).   The PPP ranges (0.01 ~ 5.05 gC m$^{-2}$ day$^{-1}$)
were similar to previous $^{14}$C measurement PP values of between 0.04 ~ 5.9 gC m$^{-2}$ day$^{-1}$.

Based on our model calculation, which assumes all the nutrients are available for

production, the PPP showed maxima at all times in sub-region A (near the Mississippi river) and
minima in sub-region B (between the Mississippi and Atchafalaya River), except for MCH M2
in June 2004, when sub-region C had the lowest PPP (Figure 6a).   The high values in sub-
region A are due largely to underutilization of nutrients in regions of high turbidity.   As the
water flows west under the influence of the Coriolis effect, PPP is expected to decrease as a
result of declining nutrient concentrations because of dilution and nutrient uptake during
biological production while the water flows to sub-region B.   In sub-region C, MCH M4
(March 2005) had the highest PPP among the all MCH cruises.   This probably depended on
high nutrient concentrations being present during the winter period, when the region was affected
by Atchafalaya River nutrient input.

*Model scenarios in the Gulf of Mexico (GOM)*

We tested the sensitivity of the model to changes in input/output parameters such as

increasing AN-D and decreasing riverine N input. Assuming the model is robust, we

investigated three model scenarios based on the nutrient distributions seen during the MCH1

cruise (note that using data from other cruises gives very similar results). In the first scenario,

we cut riverine N input 60% and increased the AN-D input by a factor of two based on

increasing N emission predictions (Duce et al., 2008; He et al., 2010; Kanakidou et al., 2016;

Kim (T) et al., 2011; Lawrence et al., 2000; Paerl et al., 2002). In the second scenario, we

doubled the amount of AN-D as in scenario 1 and decreased riverine N input by 30% based on

the hypoxia management plan goal (Gulf Hypoxia Action Plan Report, 2001, 2008; Rabalais et al.

2009). In the third scenario, we increased riverine N input by 20%, assuming the failure of the

hypoxia management plan, while we set the AN-D amount equal with the first and second

scenarios. Based on our N-mass balance model calculation and model scenarios, we can

initially estimate carbon fluxes from our PPP rate, and, using the Redfield carbon to oxygen

stoichiometry ratio (106:138), the overall oxygen balance within the coastal GOM (Table 4).

As can be seen in the scenario results for MCH M1 data (Table 4), the riverine N input

source is still the major controlling factor in the coastal GOM region even when its contribution

is greatly reduced and the AN-D source is doubled. For instance, if we fail to reduce riverine N

input in the future (scenario 3), the potential carbon fluxes will increase by 17% relative to

current conditions. In contrast, the AN-D input source only increased to a maximum of 5% of

the total input term and this indicates that AN-D input is still a minor factor in the GOM. If the

production is increased, overall oxygen demand will also be increased. The MCH M1 scenario

result indicated that the overall oxygen demand would increase approximately 21% if we fail to

reduce riverine N input, likely increasing considerably the area of the hypoxia.


*An N-mass balance model in the Coastal Sea off Korea (CSK)*
As we have done in the GOM, we used our N-mass balance model to estimate the PPP in
the CSK and define the three different zones (Figure 7).   Similar to the GOM region, the PPP
rates were highest near the river mouth, and we set the boundaries of each zone based on our N-
mass balance model results.   Based on nutrient data, as was done for the GOM, we defined the
brown zone as having a PPP rate above 1.5 gC m$^{-2}$ day$^{-1}$ because of the increased N sources from
the river, AN-D, and the sediment flux.   We defined the green zone as having PPP rates
between 0.3 to 1.5 gC m$^{-2}$ day$^{-1}$ and the blue zone as having rates of less than 0.3 gC m$^{-2}$ day$^{-1}$.
The seasonal results shown in Figures 7a and b show that the boundaries of the three
zones above and below the pycnocline layer were roughly consistent with the main change
coming in summer (August), which is the wet season and sees the highest river discharge.   The
large size of the green zone in all seasons suggests that AN-D is consistently adding extra
nitrogen to the surface ocean along with the riverine N input.   This is supported by the fact that
the PPP in the blue zone is an order of magnitude higher than for the GOM.   Around 90% of the
grid cells in the CSK are in the same zones above and below the pycnocline (Figure 7 a and b)
during all four cruises; however, in the GOM (Figure 6 a and b) this was found for fewer than
half of the grid cells.   This is probably due to the difference in freshwater discharge rate in the
two regions, which leads to a much larger stratified area in the GOM than in the CSK.
One question that has not been investigated is the temperature dependence of primary
productivity in the two areas.   While the GOM is temperate throughout the year, winter
temperatures in the CSK fall to ~5°C.   However, according to the ocean color remote sensing
images from near the CSK river mouth reported by Son et al., (2005), primary production in the
CSK does not appear to be strongly affected by temperature.   The PPP results of our model (0.2
to 2.2 gC m$^{-2}$ day$^{-1}$) agreed with their ocean color remote sensing results (0.4 to 1.6 gC m$^{-2}$ day$^{-1}$)
in the CSK.   Also, during all seasons, the Keum River consistently supplies high amounts of
DIN (average: < 60 μM) (Lim et al., 2008) to the coastal zone (especially close to the Keum
mouth).   We believe, therefore, that the higher value of PPP in winter near the Keum mouth
(brown zone in figure 7a), is reasonable.

The AN-D input source comes mainly from the Chinese side of the East China Sea (ECS)

and this affects the boundaries of the green and blue zones above the pycnocline as it is
deposited uniformly across the region.   There is also nutrient input from offshore, as the Yellow
Sea Bottom Cold Water Mass can up-well during the mixing process and is assumed to supply
additional nutrients to the outer shelf (Lim et al., 2008).

*Model scenarios in Mid-Western Coastal Sea off Korea (CSK)*

AN-D is currently considerably more important (by approximately an order of magnitude)

in the CSK than in the GOM), and it is anticipated that AN-D will likely be a major controlling
factor here in the future (Duce et al., 2008; He et al., 2010; Kim (T) et al., 2011; Lawrence et al.,
2000; Paerl et al., 2002).   Because of the lack of research on potential hypoxia scenarios in
Korea, we used the same three scenarios in the CSK as were used for the GOM.   Similar to
GOM results, riverine N input remains the major controlling factor; however, in this area, the
AN-D source is more critical than in the GOM region (Table 5).   The AN-D input source
increased from 20% to 47% of the total input under scenario 1, while based on our scenario 3
results, increases in the AN-D input source and riverine N input together will affect biological
production by increasing carbon fluxes up to 25% and oxygen demand up to 32% if we fail to
reduce N input in future (Table 5).

**Discussion**

Most previous model studies in the GOM were focused on predicting the hypoxia area

(Bierman et al., 1994; Fennel et al., 2011, 2013; Justic et al., 1996, 2002, 2003; Scavia et al.,
2004). For example, Justic et al., (1996; 2003) used a two-layer model incorporating vertical
oxygen data, from one station (LUMCON station C6; 28.867°N, 90.483°W), to predict the size
of the hypoxia area.   Similarly, Fennel et al. (2011; 2013) used her more complex simulation
model, which included oxygen concentration as well as a plankton model from Fasham et al.
(1990), to predict the size of the hypoxia region in the GOM.   Our N-mass balance model, in
contrast, uses historical data from the LATEX shelf to estimate potential carbon fluxes in the
GOM, and calculate the overall oxygen demand from those carbon fluxes.   While this affects
the total area subject to hypoxia it does not estimate the size of the hypoxic zone.

In contrast to our model, traditional predictive models have also ignored different

nitrogen input sources such as AN-D and SGD.   While this is probably reasonable on the
Texas-Louisiana shelf, where riverine inputs dominate, it may not apply in other coastal regions.
As a result, model studies in this region have concluded that reducing riverine N input is the only
solution to decrease the size of the hypoxia area in the GOM (Gulf Hypoxia Action Plan Report,
2001, 2008; Rabalais et al. 2009; Scavia et al., 2013).   According to our model results, AN-D is
still a minor controlling factor in the GOM; however, in the CSK, the AN-D contributed more to
the total nitrogen budget and may be a major controlling factor in the future.   This indicates that
AN-D should be considered as another input term for nutrient managements, especially in Asia
or in other regions where high concentrations are expected. Similarly, nitrogen input from
either sediment fluxes or groundwater also need to be considered.
Our zonal boundaries can be compared with the results of Lahiry (2007), who used
salinity to define the edges of each zone for the three cruises MCH M1, M2, and M3 (Figure 8)
and defined the edges of the RC02 zones in the coastal GOM based solely on salinity. Her
limited simulation results indicated similar patterns to our model based on DIN concentration
near the Mississippi River mouth (e.g., during MCH M1, M2, and M3). Mixing was more
conservative in this region than further west because the low salinity water with high nutrients
was less diluted with offshore water.
Away from the MR in sub-regions B and C, however, her results gave very different
boundaries for the three zones compared with our results (Figure 8). In particular, the results
near the Atchafalaya River were very different (compare Figures 6 and 8). For example, our
data showed only green and blue zones off Atchafalaya Bay during MCH M1, with no brown
zone. Similarly, the extent of the blue zones in sub-region C during MCH M2 and M3 is also
very different. We believe that our N-model based classification can cover more complex
biological processes than the Lahiry (2007) method, which considers only advection and mixing
and that our N-model is a more sensible way to look at biological processes in the GOM.
Our results also agree with previous studies that demonstrated that both the GOM and
CSK regions are N-limited for most of the year (Kim (G) et al., 2011; Turner and Rabalais,
2013). This compares with the results of Sylvan et al., (2007), who reported that the coastal
GOM could be P-limited in the MR delta mouth area where our brown zone is located, while
RC02 suggested light-limitation rather than N- or P-limitation. However, this P-limited
condition appears to occur when N concentrations are very high. In particular, the N/P ratios in
the both the GOM and CSK during our sampling were less than 16, indicating that both regions
were N-limited, although a few stations in the brown zone near the MR river area had ratios of
between 16 and 18 (Figure 9).    These higher N to P ratios may result from the high sediment
turbidity causing light-limited conditions in this zone near the river mouth (Rowe and Chapman,

2002).

It should be remembered, however, that the arithmetic N:P value per se is unimportant in

determining nutrient limitation.    As long as both nutrients can be measured, it is theoretically
possible for phytoplankton to continue to grow.    The MARS has generally such an excess of N
relative to P that N:P ratios >>16 can be expected as P concentrations fall, but this does not
necessarily mean that productivity is limited, and we never found P concentrations of zero in any
of our sub-regions; the lowest P concentration measured during all cruises in the GOM and CSK
was 0.2 µM.

Both the GOM and CSK regions receive nitrogen inputs from AN-D, rivers, and benthic

fluxes.    These different nitrogen input sources control coastal productivity, and this may reflect
the different nitrogen cycling in the two regions.    In the GOM, the riverine N input source
consistently dominates coastal productivity and eutrophication, while, in the CSK, AN-D is also
becoming a critical controlling factor.    In the CSK, however, there is strong tidal mixing of
freshwater from the Keum River and/or Gyunggi Bay with nearby coastal water, which results in
a tidal front along the offshore region and off the Taean Peninsula during spring and summer.
It is this physical mixing that mostly controls the spatial distribution patterns of nutrients and
salinity here, particularly below the pycnocline (Lim et al., 2008).    The brown zone in the upper
layer in the CSK (August 2008) changed to a green zone region below the pycnocline layer as a
result of the strong coastal tidal mixing.
RC02 considered their model to be theoretical.  In the brown zone, close to the river
mouth, they assumed turbidity leads to light-limited conditions.   Their results agree well with
measured [14]C PP numbers from Quigg et al. (2011) who found the lowest integrated PP is near
the MR delta mouth.  However, our N-mass balance model did not consider light limitation and
therefore PPP in the brown zone is high.  Such good agreement suggests that our model can be
applied to a wide region, while [14]C measurements are typically conducted at a few specific points,
as long as such limitations are taken into account.
In the CSK, most previous production studies focused on inshore areas such as estuaries
or rivers. Our research focused for the first time on the coastal ocean off Korea.  Our results
suggest that diverse nitrogen sources need to be recognized as potential issues for future nutrient
management concerned with hypoxia, eutrophication, or other environmental issues. The
agreement between our results and the pattern of production based on satellite-sensing in the
CSK (Son et al., 2005), suggests that our model is reasonable.
The results of our changing scenarios represent how the biological processes in these
coastal regions may vary as individual nutrient sources change.   While our model cannot
predict the area of the hypoxic zone, we can investigate the effects of potential flux changes of
each factor, such as AN-D, riverine input, or benthic fluxes, and calculate the effects of changes
in each on PPP and on the overall oxygen balance for the region.   We have only considered
different input terms of our N-mass balance model; output terms such as water mixing rates and
the residence time for each box need more detailed study in future work to calculate more
realistic production changes in each box.

**Conclusion**
The model suggests that the three zone theory of RC02 can be applied not only in the
northern GOM but also in the CSK region and that three zones can be distinguished based on
their nutrient concentration.    As a result, we believe that using our N-mass balance model to
separate different zones based on RC02 may be appropriate not only for large-scale regions like
the GOM and CSK but also at small scales such as river or estuary systems.    The model also
estimates potential primary production and carbon flux based on the inclusion of AN-D data that
have not been considered previously (e.g. Bierman et al., 1994; Kim (T) et al., 2011).    Our
results agree well with previous $^{14}$C measurements in the GOM (Quigg et al., 2011) and ocean
color remote sensing in the CSK (Son et al., 2005).
Based on CSK cruise data results, we can initially determine where the three different
zones are in the CSK.    We evaluated our model and tested its sensitivity based on three
different scenarios.    Through our scenario results, we assume that the AN-D is a considerable
factor in the CSK as well as the riverine N input from the Keum river.    Reducing nutrient input
from the river is critical for hypoxia management policy (Gulf Hypoxia Action Plan Report,
2001, 2008; Rabalais et al. 2009).    In addition, these model scenarios will be helpful in future
coastal nutrient management or hypoxia management studies in the CSK, especially as AN-D
sources become more important.

**Acknowledgements**
The authors would like to thank to the captain and crew of the R/V Gyre, R/V Pelican,
and R/V Manta along with the many marine technicians and students who participated in the
cruises.    This research was made possible by grant SA 12-09/GoMRI-006 to the Gulf Integrated
Spill Consortium from the Gulf of Mexico Research Initiative and by grants NA03NOS4780039,
NA06NOS4780198, and NA09NOS4780208 from NOAA.  Hydrographic and dissolved
nutrients data used in this study from the Texas-Louisiana Shelf from years 2004-2009 are
available from NOAA NCEI (accession-ID 0088164).

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

Preliminary Mass Balance Model of Primary Productivity and Dissolved Oxygen in the
Mississippi River Plume/Inner Gulf Shelf Region. Estuaries., 17(4), 886-899, 1994.

Bode, A., and Dortch, Q.: Uptake and regeneration of inorganic nitrogen in coastal waters
influenced by the Mississippi River: spatial and seasonal variations. Journal of Plankton
Resources., 18, 2251-2268, 1996.

Castro, M. S., Driscoll, C. T., Jordan, T. W., Reay, W. G., Boynton, W. R., Seitzinger, S. P.,
Styles, R. V., and Cable, J. E.: Contribution of atmospheric deposition to the total
nitrogen loads to thirty-four estuaries on the Atlantic and Gulf coasts of the United States,
in: Coastal and Estuarine Studies-Nitrogen Loading in Coastal Water Bodies: An
Atmospheric Perspective, edited by: Valigura, R. A., Alexander, R. B., Castro, M. S.,
Meyers, T. P., Paerl, H. W., Stacey, P. E., Turner, R. E., American Geophysical Union,
Washington, D.C., USA, 77-106, 2001.

Castro, M. S., and Driscoll, C. T.: Atmospheric nitrogen deposition to estuaries in the mid-
Atlantic and northeastern United States. Environmental science & technology., 36(15),
3242-9, 2002.

Cho, K. R., Reid, O., and Nowlin, Jr W. D.: Objectively mapped stream function fields on the
Texas-Louisana shelf based on 32 months of moored current meter data. Journal of
Geophysics Research., 103(C5), 10377-10390, 1998.
Choi, H. Y., Lee, S. H., and Oh, I. S.: Quantitative Analysis of the Thermal Front in the Mid-
Eastern Coastal Area of the Yellow Sea. Journal of the Korean Society of Oceanography
[The Sea]., 3, 1-8, 1998.
Choi, H. Y., Lee, S. H., and Yoo, K. Y.: Salinity Distribution in the Mid-eastern Yellow Sea
during the High Discharge from the Keum River Weir.   Journal of the Korean Society of
Oceanography [The Sea]., 4, 1-9, 1999.
Christensen, P. B., Rysgaard, S., Sloth, N. P., Dalsgaard, T., and Schwaerter, S.: Sediment
mineralization, nutrient fluxes, denitrification and dissimilatory nitrate reduction to
ammonium in an estuarine fjord with sea cage trout farms. Aquatic Microbial Ecology.,
21, 73-84, 2000.
Cornell, S., Rendell, A., and Jickells, T.: Atmospheric inputs of dissolved organic nitrogen to the
oceans. Nature., 376, 243-246, 1995.
Dagg, M. J., and Breed, G. A.: Biological effects of Mississippi River nitrogen on the northern
Gulf of Mexico-a review and synthesis.   Journal of Marine Systems., 43, 133-152, 2003.
Dagg, M. J., Ammerman, J. W., AMON, R. M. W., Gardner, W. S., Green, R. E., Lohrenz, S. E.:
A review of water column processes influencing hypoxia in the northern Gulf of Mexico.
Estuaries Coasts., 30, 735-752, 2007.
Dalsgaard, T.: Benthic primary production and nutrient cycling in sediments with benthic
microalgae and transient accumulation of macroalgae. Limnology and Oceanography.,
48(6), 2138-2150, 2003.
De Boer, A. M., Watson, A. J., Edwards, N. R., and Oliver, K. I. C.: A multi-variable box model
approach to the soft tissue carbon pump. Climate of the past., 6, 827-841, 2010.
Diaz, R. J., and Rosenberg, R.: Marine benthic hypoxia: A review of its ecological effects and
the behavioural responses of benthic macrofauna. Oceanography Marine Biology. Ann.
Rev., 33, 245-303, 1995.
Diaz, R. J., and Rosenberg, R.: Spreading dead zones and consequences for marine ecosystems.
Science., 321(5891), 926-9, 2008.
DiMarco, S. F., and Zimmerle, H. M.: MCH Atlas: Oceanographic Observations of the
Mechanisms Controlling Hypoxia Project. Texas A&M University, Texas Sea Grant,
College Station, TX. Publication TAMU-SG- 17-601,350. ISBN 978-0-692-87961-0,
2017.

643 DiMarco, S. F., Chapman, P., Walker, N., and Hetland, R. D.: Does local topography control
644    hypoxia of the Texas-Louisiana Shelf? Journal of Marine Systems., 80(1-2), 25-35, 2010.

646 Dodds, W. K., and Smith, V. H.: Nitrogen, phosphorus, and eutrophication in streams. Inland
647    Waters., 6, 155-164, 2016.

649 Doney, S. C., Mahowald, N., Lima, L., Feely, R. A., Mackenzie, F. T., Lamarque, J. F., and
650    Rasch, P. J.: Impact of anthropogenic atmospheric nitrogen and sulfur deposition on
651    ocean acidification and the inorganic carbon system. Proceedings of the National
652    Academy of Science., 104, 14580-14585, 2007.

654 Dortch, Q., and Whitledge, T. E.: Does nitrogen or silicon limit phytoplankton in the Mississippi
655    River plume and nearby regions? Continental Shelf Research., 12, 1293-1309, 1992.

657 Duce, R. A., LaRoche, J., Altierl, K., Arrigo, K. R., Baker, A. R., Capone, D. G., Cornell, S.,
658    Dentener, F., Galloway, J., Ganeshram, R. S., Geider, R. J., Jickells, T., Kuypers, M. M.,
659    Langlois, R., Liss, P. S., Liu, S. M., Middelburg, J. J., Moore, C. M., Nickovic, S.,
660    Oschlies, A., Pedersen, T., Prospero, J., Schlitzer, R., Seitzinger, S., Sorensen, L. L.,
661    Uematsu, M., Ulloa, O., Voss, M., Ward, B., and Zamora, L.: Impacts of Atmospheric
662    Anthropogenic Nitrogen on the Open Ocean. Science., 320, 893-897, 2008.

664 Fasham, M. J. R., Ducklow, H. W., and Mckelvie, S. M.: A nitrogen-based model of plankton
665    dynamics in the oceanic mixed layer. Journal of Marine research., 48, 591-639, 1990.

667 Feng, Y., DiMarco, S. F., and Jackson, G. A.: Relative role of wind forcing and riverine nutrient
668    input on the extent of hypoxia in the northern Gulf of Mexico. Geophysical Research
669    Letters., 39, L09601, 2012.

671 Feng, Y., Fennel, K., Jackson, G. A., DiMarco, S. F., and Hetland, R. D.: A model study of the
672    response of hypoxia to upwelling-favorable wind on the northern Gulf of Mexico shelf.
673    Journal of Marine Systems., 131, 63-73, 2014.

675 Fennel, K., Wilkin, J., Levin, J., Moisan, J., O'Reilly, J., and Haidvogel, D.: Nitrogen cycling in
676    the Middle Atlantic Bight: Results from a three-dimensional model and implications for
677    the North Atlantic nitrogen budget. Global Biogeochemical cycles., 20, GB3007,
678    doi:10.1029/2005G, 2006.

680 Fennel, K., Hetland, R. D., Feng, Y., and DiMarco, S. F.: A coupled physical-biological model
681    of the Northern Gulf of Mexico shelf: model description, validation and analysis of
682    phytoplankton variability. Biogeosciences., 8, 1881-1899, 2011.

684 Fennel, K., Hu, J., Laurent, A., Marta-Almeida, M., and Hetland, D. R.: Sensitivity of hypoxia
685    predictions for the northern Gulf of Mexico to sediment oxygen consumption and model
686    nesting. Journal of Geophysical Research: Oceans., 118, 990-1002, 2013.

Forrest, D. R., Hetlandl R. D., and DiMarco, S. F.: Multivariable statistical regression models of
the areal extent of hypoxia over the Texas–Louisiana continental shelf. Environmental
Research Letters., 6, 045002, 2011.

Goolsby, D. A. Mississippi basin nitrogen flux believed to cause Gulf hypoxia., EOS
Transactions 2000:29–321, 2000.

Green, R. E., Gould, Jr. R. W., and Ko, D. S.: Statistical models for sediment/detritus and
dissolved absorption coefficients in coastal waters of the northern Gulf of Mexico.
Continental Shelf Research., 28(10), 1273-1285, 2008.

Hagy, J. D., Sanford, L. P., and Boynton. W. R.: Estimation of net physical transport and
hydraulic residence times for a coastal plain estuary using box models., Estuaries 23:328–
340. doi:10.2307/1353325, 2000.

He, C. H., Wang, X., Liu, X., Fangmeler, A., Christie, P., and Zhang, F.: Nitrogen deposition and
its contribution to nutrient inputs to intensively managed agricultural ecosystems.
Ecological Application., 20(1), 80-90, 2010.

Hetland, R. D., and DiMarco, S. F.:  How does the character of oxygen demand control the
structure of hypoxia on the Texas-Louisiana continental shelf?  Journal of Marine
Systems., 70, 49-62, 2008.

Howarth, R. W., and Marino, R.: Nitrogen as the limiting nutrient for eutrophication in coastal
marine ecosystems: Evolving views over three decades. Limnology and Oceanography.,
51(1), 364-376, 2006.

Jacob, G. A., Hur, H. B., and Riedlinger, S. K.: Yellow and East China Seas response to winds
and currents. Journals of Geophysical Research., 105 (21), 947-968, 2000.

Justic, D., Rabalais, N. N., and Turner, R. E.: Effects of climate change on hypoxia in coastal
waters; A doubled CO2 scenario for the northern Gulf of Mexico.  Limnology and
Oceanography., 41(5), 992-1003, 1996.

Justic, D., Rabalais, N. N., and Turner, R. E.: Modeling the impacts of decadal changes in
riverine nutrient fluxes on coastal eutrophication near the Mississippi River Delta.
Ecological Modelling., 152, 33-46, 2002.

Justic, D., Rabalais, N. N., and Turner, R. E.: Simulated responses of the Gulf of Mexico
hypoxia to variations in climate and anthropogenic nutrient loading. Journal of Marine
Systems., 42, 115-126, 2003.

Kanakidou, M., Myriokefalitakis, S., Daskalakis, N., and Fanourgakis, G.: Past, Present, and
Future Atmospheric Nitrogen Deposition. American Meteorological Society. Journal of
the Atmospheric Sciences., 73(5), 2039-2047, 2016.

Kim, G., Kim, J. S., and Hwang, D. W.: Submarine groundwater discharge from oceanic islands
standing in oligotrophic oceans: Implications for global production and organic carbon
fluxes. Limnology and Oceanography., 56(2), 673-682, 2011.
Kim, J. S., Lee, M. J., Kim, J., and Kim, G.: Measurement of Temporal and Horizontal
Variations in $^{222}$Rn Activity in Estuarine Waters for Tracing Groundwater Inputs.
Ocean Science Journal., 45(4), 197-202, 2010.
Kim, J. S.: Implications of different nitrogen input sources for primary production and carbon
flux estimates in coastal waters. Texas A&M University. Ph.D. Dissertation., 2018.
Kim, J. Y., Ghim, Y. S., Lee, S. B., Moon, K. C., Shim, S. G., Bae, G. N., and Yoon, S. C.:
Atmospheric Deposition of Nitrogen and Sulfur in the Yellow Sea Region: Significance
of Long-Range Transport in East Asia. Water, Air, and Soil Pollution., 205, 259-272,
2010.
Kim, T. W., Lee, K., Najjar, R. G., Jeong, H. D., and Jeong, H. J.: Increasing N abundance in the
northwestern Pacific Ocean due to atmospheric nitrogen deposition. Science., 334, 505-
509, 2011.
Lahiry, S.: Relationships between nutrients and dissolved oxygen concentrations on the Texas-
Louisiana shelf during spring-summer of 2004. Texas A & M University. MS. Thesis.,
2007.
Lamarque, J. F., Dentener, F., McConnell. J., Ro. C. U., Shaw. M., Vet. R., Bergmann. D.,
Cameron-Smith. P., Dalsoren. S., Doherty. R., Faluvegi. G., Ghan. S. J., Josse. B., Lee. Y.
H., MacKenzie. I. A., Plummer. D., Shindell. D. T., Skeie. R. B., Stevenson. D. S., Strode.
S., Zeng. G., Curran. M., Dahl-Jensen. D., Das. S., Fritzsche. D., and Nolan. M.: Multi-
model mean nitrogen and sulfur deposition from the Atmospheric Chemistry and Climate
Model Intercomparison Project (ACCMIP): evaluation of historical and projected future
changes. Atmospheric Chemistry and Physics., 13, 7997-8018, 2013.
Lawrence, G. B., Goolsby, D. A., Battaglin, W. A., and Stensland, G. J.: Atmospheric nitrogen in
the Mississippi River Basin-emissions, deposition and transport. Science of The Total
Environment., 248(2-3), 87-100, 2000.
Laurent, A., Fennel, K., Hu, J., and Hetland, R. D.: Simulating the effects of phosphorus
limitation in the Mississippi and Atchafalaya River plumes. Biogeosciences., 9, 4797-
4723, 2012.
Lee, J. S., Kim, K. H., Shim, J. H., Han, J. H., Choi, Y. H., and Khang, B. J.: Massive
sedimentation of fine sediment with organic matter and enhanced benthic-pelagic
coupling by an artificial dyke in semi-enclosed Chonsu Bay, Korea. Marine Pollution
Bulletin., 64, 153-163, 2012.

Lim, D., Kang, M. R., Jang, P. G., Kim, S. Y., Jung, H. S., Kang, Y. S., and Kang, U. S.: Water
Quality Characteristics Along Mid-Western Coastal Area of Korea. Ocean and Polar
Research., 30(4), 379-399, 2008.
Liu, S. M., Zhang, J., Chen, S. Z., Chen, H. T., Hong, G. H., Wei, H., and Wu, Q. M.: Inventory
of nutrient compounds in the Yellow Sea. Continental Shelf Research., 23, 1161-1174,
2003.
Lehrter, J. C., Beddick Jr., D. L., Devereux, R., Yates, D. F., and Murrell. M. C.: Sediment-water
fluxes of dissolved inorganic carbon, $O_2$, nutrients, and $N_2$ from the hypoxic region of
the Louisiana continental shelf. Biogeochemistry, 109, 233–252, 2012.
Lohrenz, S. E., Wiesenburg, D. A., Arnone, R. A., and Chen, X. G.: What controls primary
production in the Gulf of Mexico? In: Sherman K, Kumpf H, Steidinger K (ed) The Gulf
of Mexico Large Marine Ecosystem: Assessment, sustainability and management.
Blackwell Science, Malden, MA., 151-170, 1998.
Lohrenz, S. E., Fahnenstiel, G. L., Redalje, D. G., Lang, G. A., Dagg, M. J., Whitledge, T. E.,
and Dortch, Q.: Nutrients, irradiance, and mixing as factors regulating primary
production in coastal waters impacted by the Mississippi River plume. Continental Shelf
Research., 19, 1113-1141, 1999.
Luo, X. S., Tang, A. H., Shi, K., Wu, L. H., Li, W. Q., Shi, W. Q., Shi, X. K., Erisman, J. W.,
Zhang, F. S., and Liu, X. J.: Chinese coastal seas are facing heavy atmospheric nitrogen
deposition. Environmental Research Letters., 9, 1-10, 2014.
McCarthy, M. J., Newell, S. E., Carini, S. A., and Cardner, W. S.: Denitrification dominates
sediment nitrogen removal and is enhanced by bottom-water hypoxia in the Northern
Gulf of Mexico. Estuaries and Coasts., 38, 2279-2294. 2015.
Milliman, J. D., and Meade, R. H.: World-wide delivery of river sediment to the oceans. The
Journal of Geology., 91(1), 1-21, 1983.
Mississippi River/Gulf of Mexico Watershed Nutrient Task Force.: Action Plan for Reducing,
Mitigating, and Controlling Hypoxia in the Northern Gulf of Mexico, Washington, D.C.
33 pp., 2001.
Mississippi River/Gulf of Mexico Watershed Nutrient Task Force.: Gulf Hypoxia Action Plan
2008 for Reducing, Mitigating, and Controlling Hypoxia in the Northern Gulf of Mexico
and Improving Water Quality in the Mississippi River Basin, Washington, D.C. 61 pp.,
2008.
Nipper, M., Sanchez Chavez, J. A., Tunnell, Jr. J. W.: GulfBase: Resource Database for Gulf of
Mexico Research: Corpus Christi, Texas A&M University, http://www.gulfbase.org,
2004.

Nowlin, W. D. Jr., Jochens, A. E., Reid, R. O., and DiMarco, S. F.: Texas-Louisiana Shelf
Circulation and Transport processes Study: Synthesis Report, Volume I: Technical
Report. OCS Study MMS 98-0035. U.S. Dept. of the Interior, Minerals Mgmt Service,
Gulf of Mexico OCS Region, New Orleans, LA., 502, 1998a.

Nowlin, W. D. Jr., Jochens, A. E., Reid, R. O., and DiMarco, S. F.: Texas-Louisiana Shelf
Circulation and Transport processes Study: Synthesis Report, Volume II: Appendices.
OCS Study MMS 98-0036. U.S. Dept. of the Interior, Minerals Mgmt Service, Gulf of
Mexico OCS Region, New Orleans, LA., 288, 1998b.

Nunnally, C., Quigg, A., DiMarco, S. F., Chapman, P., and Rowe, G. T.: Benthic-Pelagic
Coupling in the Gulf of Mexico Hypoxic Area: Sedimentary enhancement of hypoxic
conditions and near bottom primary production. Continental Shelf Research., 85, 143-152,
2014.

Paerl, H. W., Dennis, R. L., and Whitall, D. R.: Atmospheric Deposition of Nitrogen:
Implications for Nutrient Over-Enrichment of Coastal Waters. Estuaries., 25(4B), 677-
693, 2002.

Paerl, H. W.: Controlling Eutrophication along the Freshwater-Marine Continuum: Dual Nutrient
(N and P) Reductions are Essential. Estuaries and Coasts., 32, 593-601, 2009.

Park, M. J., Savenije, H. H. G., Cai, H., Jee, E. K., and Kim, N. H.: Progressive change of tidal
wave characteristics from the eastern Yellow Sea to the Asan Bay, a strongly convergent
bay in the west coast of Korea. Ocean Dynamics., 67, 1137-1150, 2017.

Park, Y. H.: Analysis of characteristics of Dynamic Tidal Power on the west coast of Korea.
Renewable and Sustainable Energy Reviews., 68, 461-474, 2017.

Qureshi, N. A.: The role of fecal pellets in the flux of carbon to the sea floor on a river-
influenced continental shelf subject to hypoxia. Louisiana State University. Ph.D.
Dissertation., 1995.

Quigg, A., Sylvan, J., Gustafson, A., Fisher, T., Oliver, R., Tozzi, S., and Ammerman, J.: Going
west: nutrient limitation of primary production in the northern Gulf of Mexico and the
importance of the Atchafalaya River. Aquatic Geochemistry., 17, 519-544, 2011.

Rabalais, N. N., and Smith, L. E.: The effects of bottom water hypoxia on benthic communities
of the southeastern Louisiana continental shelf. New Orleans, Louisiana, U.S. Minerals
Management Service, Gulf of Mexico OCS Region.,105, 1995.

Rabalais, N. N., and Turner, R. E.: Hypoxia in the Northern Gulf of Mexico: Description, causes
and change, pp. 1–36. In N. N. Rabalais and R. E. Turner (eds.), Coastal Hypoxia:
Consequences for Living Resources and Ecosystems. Coastal and Estuarine Studies., 58,
2001.

Rabalais, N. N., Turner, R. E., and Scavia, D.: Beyond science into policy: Gulf of Mexico
hypoxia and the Mississippi river. Bioscience., 52(2), 129-142, 2002.
Rabalais, N. N., Turner, R. E., Sen Gupta, B. K., Boesch, D. F., Chapman, P., and Murrell, M. C.:
Hypoxia in the northern Gulf of Mexico: Does the science support the plan to reduce,
mitigate, and control hypoxia? Estuaries Coastal., 30, 753-772, 2007.
Rabalais, N. N., Turner, R. E., Justic, D., Diaz, R. J.: Global change and eutrophication of coastal
waters. ICES. Journal of Marine Science., 66, 1528–1537, 2009.
Ramesh. R., Chen. Z., Cummins. V., Day. J., D'Elia. C., Dennison. B., Forbes. D. L., Glaeser. B.,
Claser. M., Clavovic. B., Kremer. H., Lange. M., Larsen. J. N., Tissier. M. Le., Newton.
A., Pelling. M., Purvaja. R., and Wolanski. E.: Land-ocean interactions in the coastal
zone: past, present & future, Anthtropocene., 12, 85-98, 2015.
Redalje, D. G., Lohrenz, S. E., and Fahnenstiel, G. L.: The relationship between primary
production and the vertical export of particulate organic matter in a river impacted coastal
ecosystem. Estuaries., 17, 829-838, 1994.
Robertson, D. M., and Saad, D. A.: SPARROW Models Used to Understand Nutrient Sources in
the Mississippi/Atchafalaya River Basin. Journal of Environmental Quality., 42, 1422-
1440, 2014.
Rowe, G. T., and Chapman, P.: Continental Shelf Hypoxia: some nagging questions. Gulf
Mexico Science., 20, 153-160, 2002.
Rowe, G. T., Kaegi, M. E. C., Morse, J. W., Boland, G. S., and Briones, E. G. E.: Sediment
community metabolism associated with continental shelf hypoxia, northern Gulf of
Mexico. Estuaries., 25(6), 1097–1106, 2002.
Scavia, D., Justic, D., and Bierman, V. J.: Reducing Hypoxia in the Gulf of Mexico: Advice
from Three Models. Estuaries., 27(3), 419-425. 2004.
Scavia, D., Evans, M. A., and Obenour, D. R.: A scenario and forecast model for Gulf of Mexico
hypoxic area and volume. Environmental Science and Technology., 47, 10423-10428,
2013.
Shou, W., Zong, H., Ding, P., and Hou, L.: A modelling approach to assess the effects of
atmospheric nitrogen deposition on the marine ecosystem in the Bohai Sea, China.
Estuarine, Coastal and Shelf Science., 208, 36-48, 2018.
Sigman, D. M., and Hain, M. P.: The Biological Productivity of the Ocean. Nature Education., 3,
1-16, 2012.

Son, S. H., Campbell, J. W., Dowell, M., Yoo, S. J., and Noh, J.: Primary production in the
Yellow Sea determined by ocean color remote sensing. Marine Ecology Progress Series.,
303, 91-103, 2005.
Sylvan, J. B., Dortch, Q., Nelson, D. M., Maier Brown, A. F., Morrison, W., Ammerman, J. W.:
Phosphorus limits phytoplankton growth on the Louisiana shelf during the period of
hypoxia formation.   Environmental Science and Technology., 40(24), 7548– 7553, 2006.
Testa, J. M., Kemp. W. M., Boynton. W. R., and Hagy. J. D: Long-term changes in water quality
and productivity in the Patuxent river estuary: 1985 to 2003. Estuaries and Coasts., 31,
1021-1037, 2008.
Thornton, D. C. O., Dong, L. F., Underwood, G. J. C., and Nedwell, D. B.: Sediment-water
inorganic nutrient exchange and nitrogen budgets in the Colne Estuary, UK. Marine
Ecology Progress Series., 337, 63-77, 2007.
Turner, R. E., and Rabalais, N. N.: Changes in the Mississippi River nutrient supply and offshore
silicate-based phytoplankton community responses, in: Changes in Fluxes in Estuaries:
Implications from Science to management Proceedings of ECSA22/ERF Symposium.
International Symposium Series, Olsen, Gredensborg, Denmark.,147-150, 1994.
Turner, R. E., Rabalais, N. N., and Justic, D.: Predicting summer hypoxia in the northern Gulf of
Mexico: riverine N, P and Si loading. Marine Pollution Bulletin., 51, 139-148, 2006.
Turner, R. E., Rabalais, N. N., and Justic, D.: Gulf of Mexico Hypoxia: Alternate States and a
Legacy. Environmental Science & Technology., 42, 2323-2327, 2008.
Turner, R. E., and Rabalais, N. N.: Nitrogen and phosphorus phytoplankton growth limitation in
the northern Gulf of Mexico. Aquatic microbial ecology., 68, 159–169, 2013.
Wade, T. L., and Sweet, S. T.: Final Report Coastal Bend Bays and Estuaries Program (CBBEP):
Atmospheric Deposition Study, Coastal Bend Bays & Estuaries Program, Corpus Christi,
Texas, USA, 48 pp, 2008.
Wang, H., Dai, M., Liu, J., Kao, S. J., Zhang, C., Cai, W. J., Wang, G., Qian, W., Zhao, M., and
Sun, Z.:  Eutrophication-Driven Hypoxia in the East China Sea off the Changjiang
Estuary. Environmental Science & Technology., 50, 2255-2263, 2016.
Yang, J. S., and Ahn, T. Y.: The analysis of the correlation between groundwater level and the
moving average of precipitation in Kuem river watershed.   The Journal of Engineering
Geology., 18, 1-6, 2008.
Zhao, Y., Zhang, L., Pan, Y., Wang, Y., Paulot, F., and Henze, D. K.: Atmospheric nitrogen
deposition to the northwestern Pacific: seasonal variation and source attribution. Atmos.
Chem. Phys., 15, 10905-10924, 2015.

**List of Figures**
Figure 8. Distribution of the three zones during cruises MCH M1-M3 based on salinity data
1008          (Lahiry, 2007). Areas shaded in three colors represent the brown, green and blue
1009          zones respectively.
Figure 9. Dissolved inorganic nitrogen (DIN) against dissolved inorganic phosphorus (DIP)
during sampling periods in the Gulf of Mexico (GOM) and Mid-western Korea (CSK).
Nearly all samples had an N:P ratio of < 16, which indicated potential N-limited
condition. At a few points near the brown zone the ratio was between 16 -18; this is
where light-limitation is expected according to RC02.

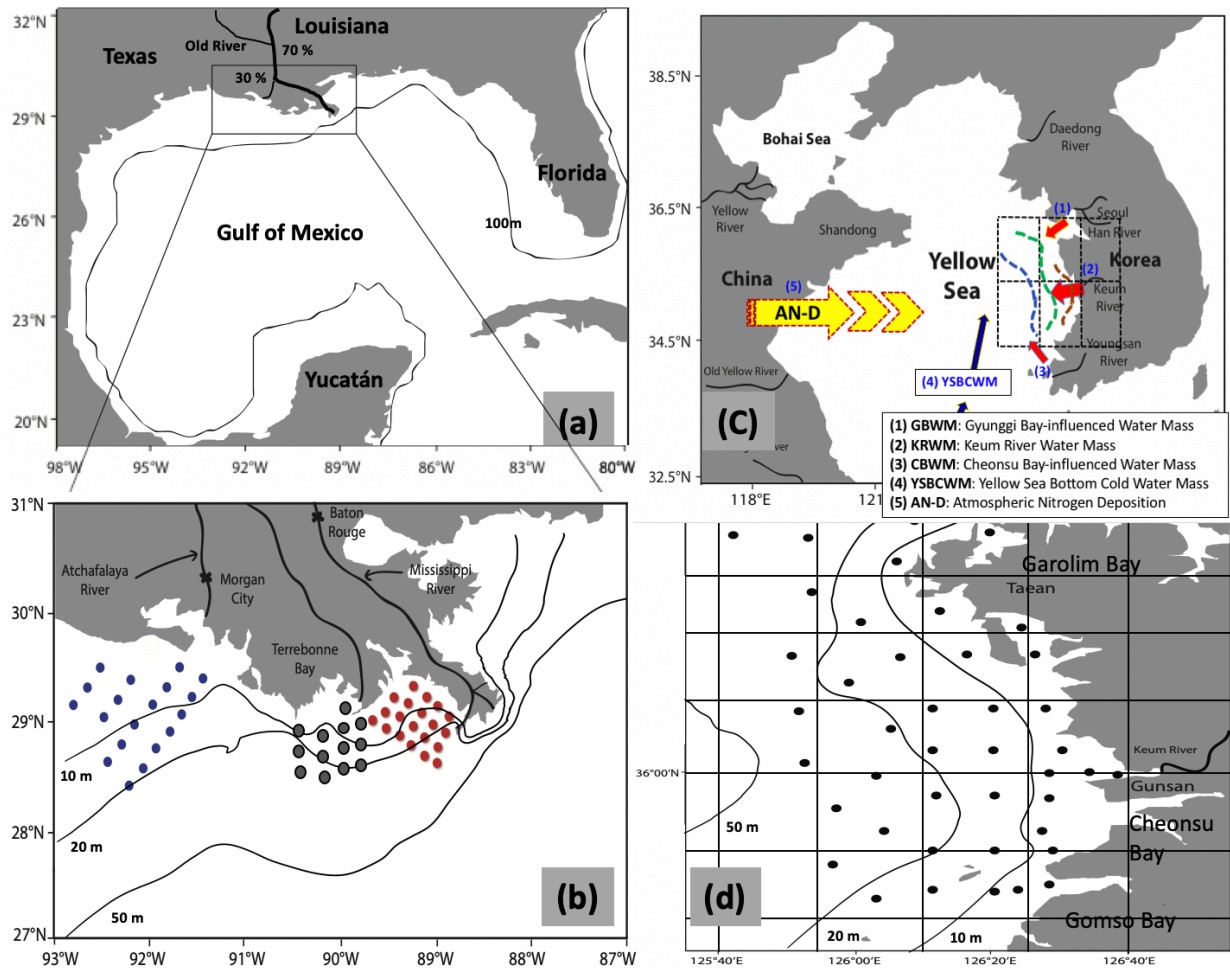

1042

**Figure 1.** Study sites and sampling areas in the Gulf of Mexico and Korea. (a) shows the sampling area within the northern Gulf of Mexico. Flow in the Mississippi/ Atchafalaya River System is split 30% to the Atchafalaya River, 70% to the Mississippi River. The box is the sampling area. (b) shows station positions from March 2005. Note that MCH project data are widely distributed across the region. Red, grey, and blue stations correspond to sub-regions A (near the Mississippi River), B (between the Mississippi and Atchafalaya), and C (near the Atchafalaya) respectively. (c) shows the sampling area off the west coast of Korea. (d) shows all of the station positions.

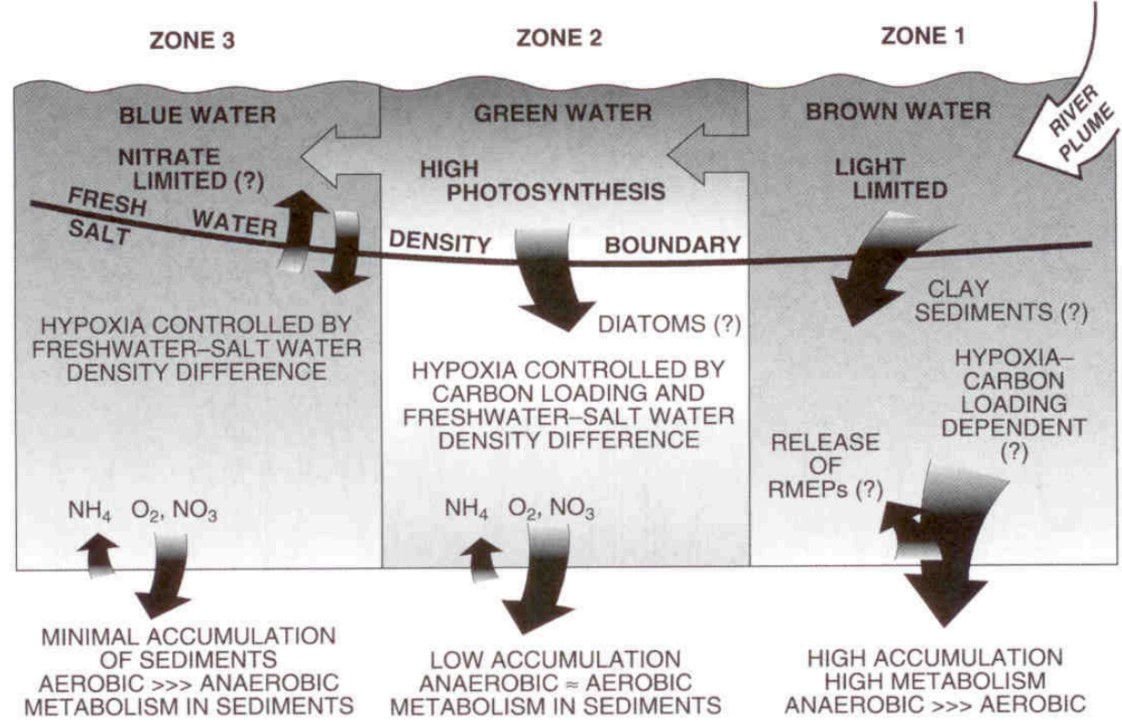


**Figure 2.** The Rowe and Chapman three zone hypothesis, which describes the physical and
biochemical processes that initiate and sustain hypoxia on the Texas-Louisiana Shelf, [Rowe and
Chapman, 2002]. RMEPs are Reduced Metabolic End Products. *Reprinted with permission of*
*Gulf of Mexico Science.*

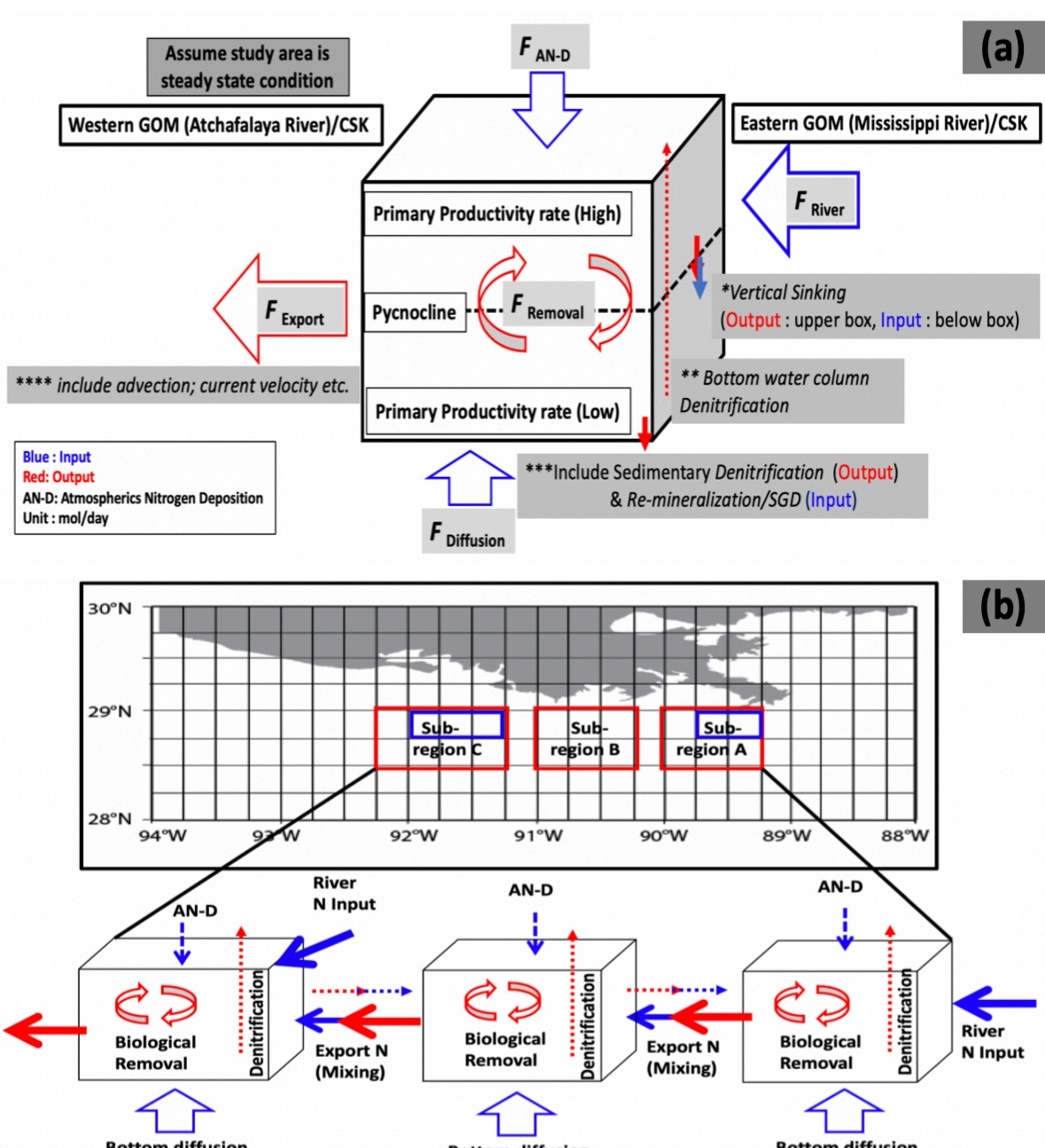


**Figure 3.** (a) Input (blue) and output (red) sources for each 0.25° box (see text for details); (b) Area of each sub-region (red) and boxes affected by direct riverine input (blue). Export N (Mixing) represents the advective transport term. The processes of biogeochemical and transport processes of both regions are the same and each in/out put factor is the same in the GOM and CSK. Note that transfer between boxes occurs in both directions alongshore and onshore/offshore and is not a one-dimensional process as suggested in the diagram.


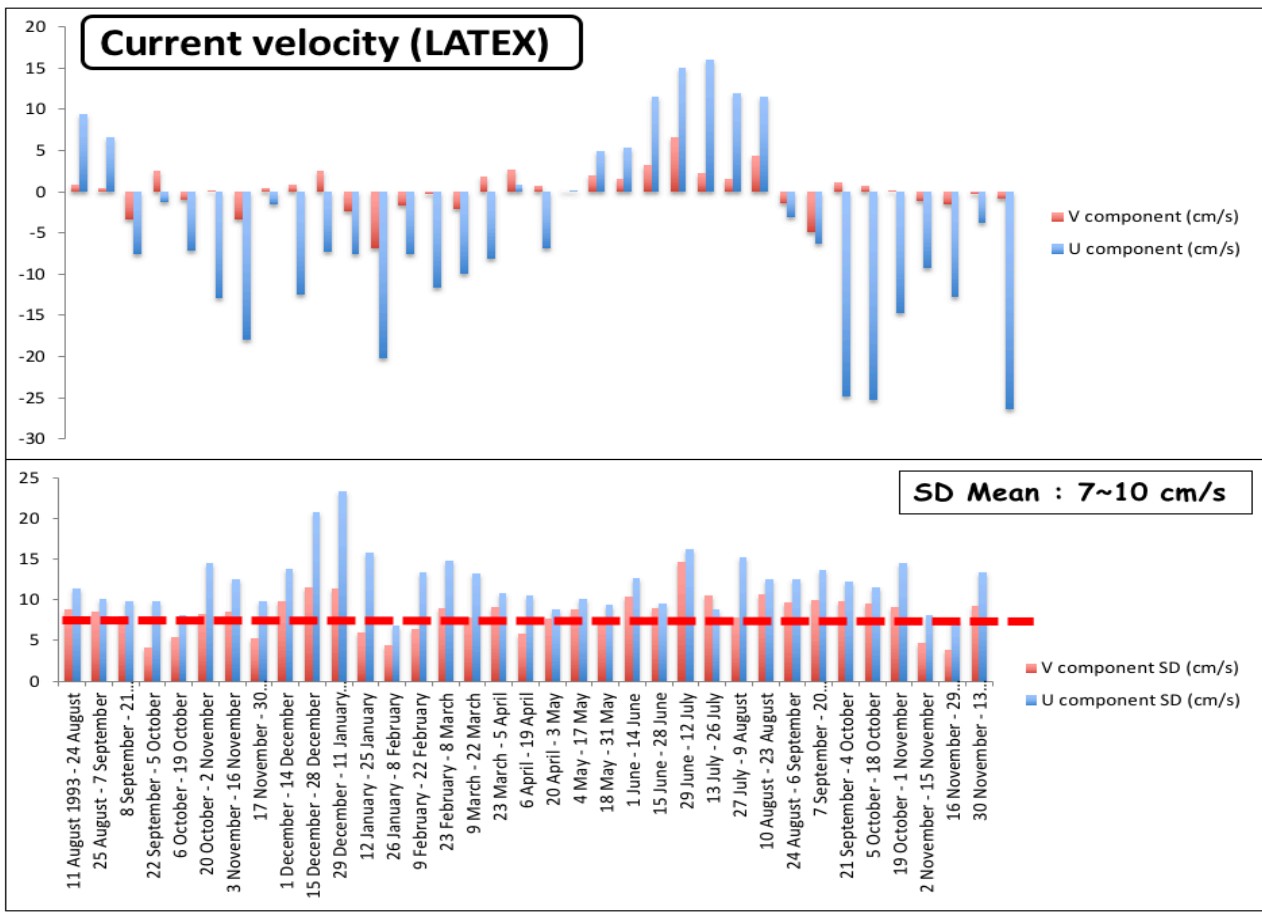

**Figure 4.** Mean ocean current velocities (a) and standard deviations (b) for biweekly periods
from August 1993 through December 1994 based on data from LATEX project. Positive values
of U show eastward flow; positive values of V show northward flow.

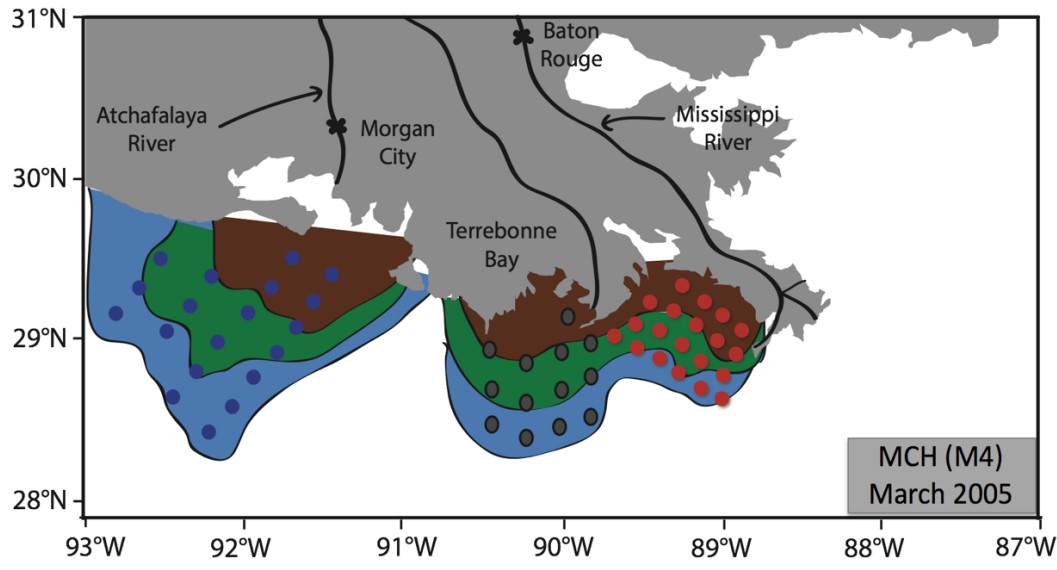


**Figure 5.** Extent of the three zones defined by RC02 based on the mean concentration of nutrient (DIN) at each station during the MCH M4
cruise in March 2005, showing their correspondence to the three sub-regions used in the box model.   Red, grey and blue stations correspond to
sub-regions A (near the Mississippi River), B (between the Mississippi and Atchafalaya), and C (near the Atchafalaya) respectively.

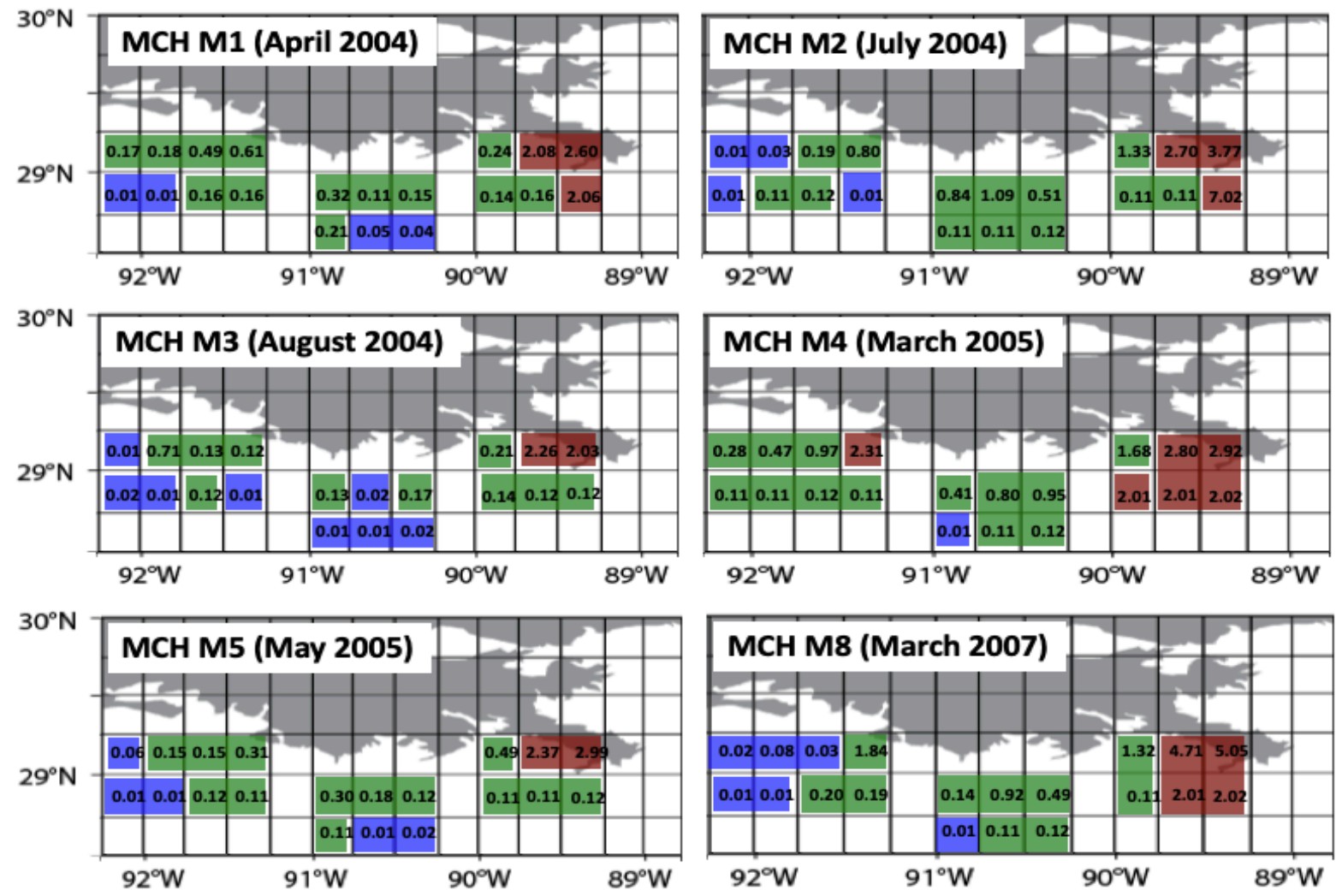


**Figure 6a.** Areal distributions of the three zones using data from above the pycnocline, based on N-mass balance model results. Colors and numbers represent boxes found in each of the three zones in terms of potential productivity (Unit: gC m$^{-2}$ day$^{-1}$).

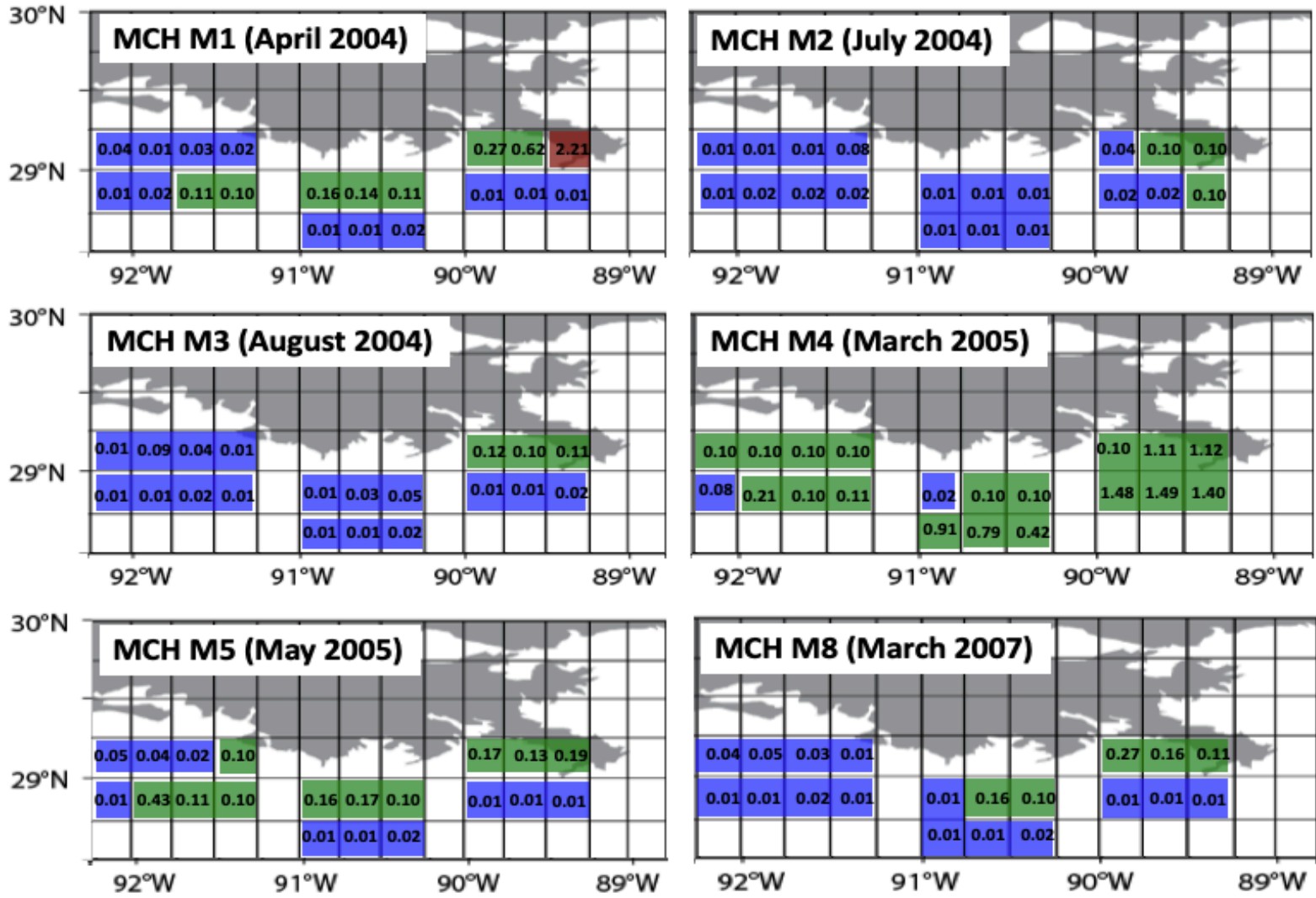


**Figure 6b.** As for 6a, using data from below the pycnocline.

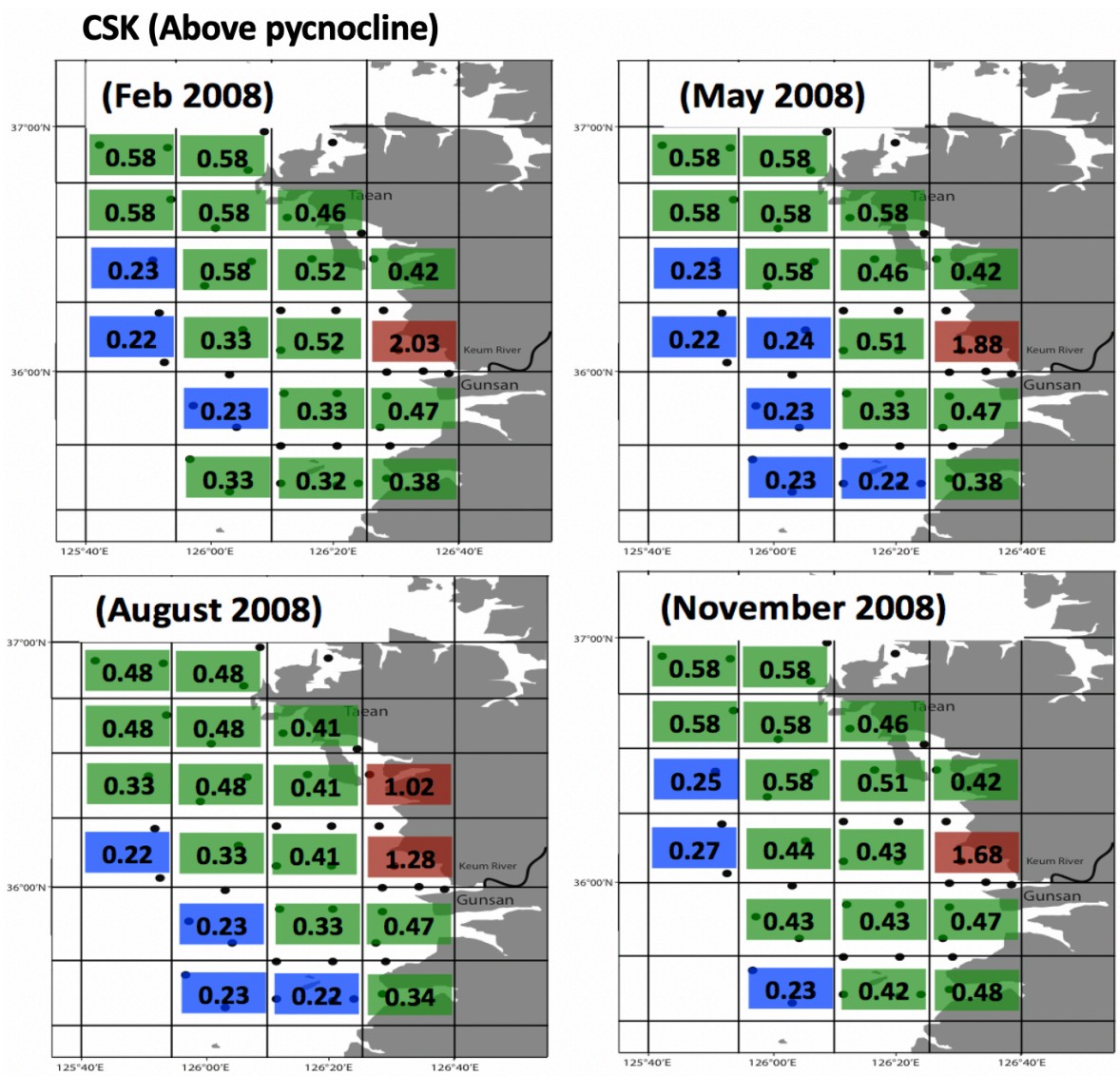


**Figure 7a.** The distribution of the three zones off Mid-western Korea (CSK) above the
pycnocline based on the RC02 hypothesis applied to the N-mass balance model.   Colors and
numbers represent boxes found in each of the three zones in terms of potential productivity
(Unit: gC m$^{-2}$ day$^{-1}$).

**CSK (Below pycnocline)**

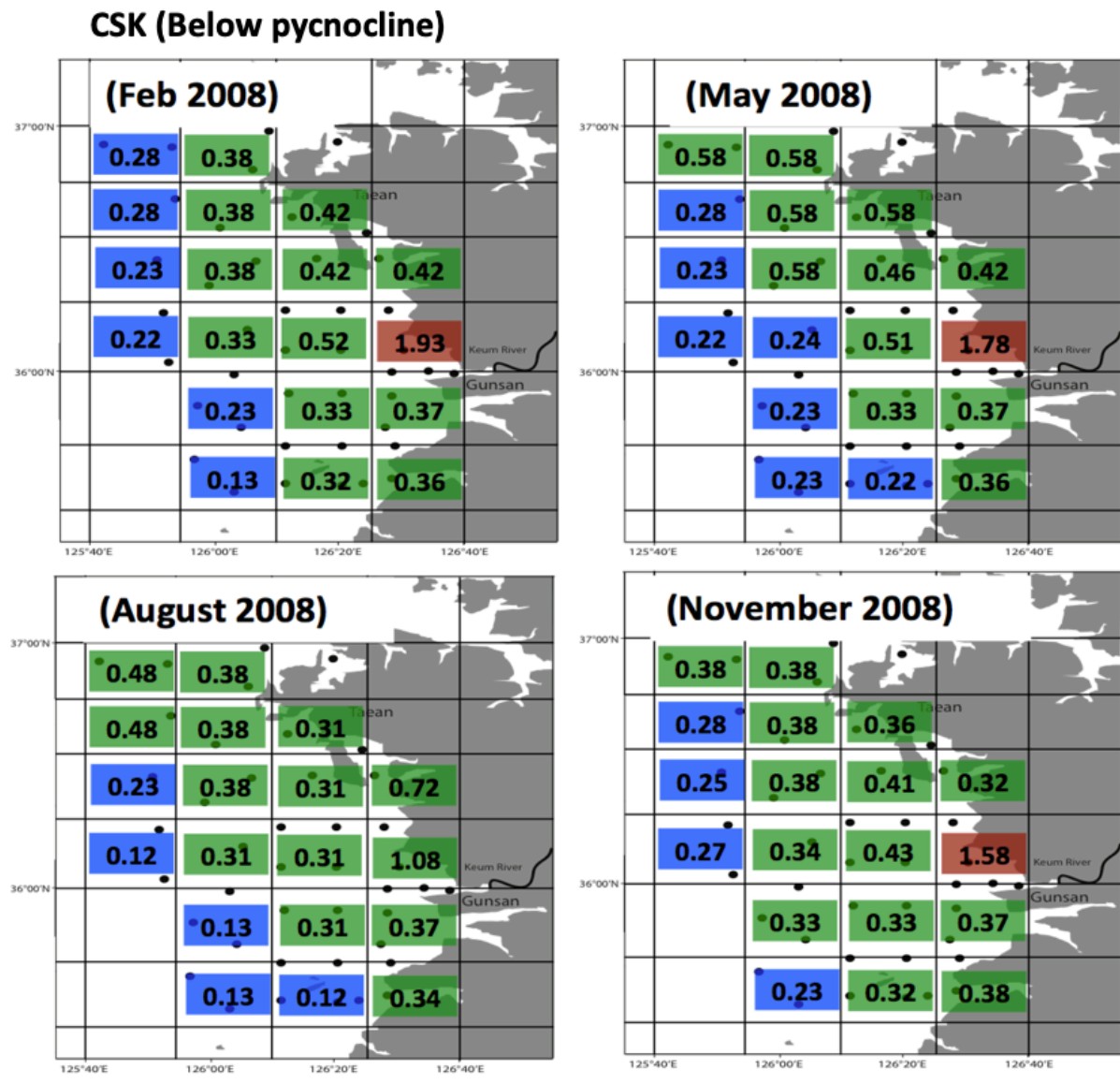


**Figure 7b.** As for 7a, using data from below the pycnocline.

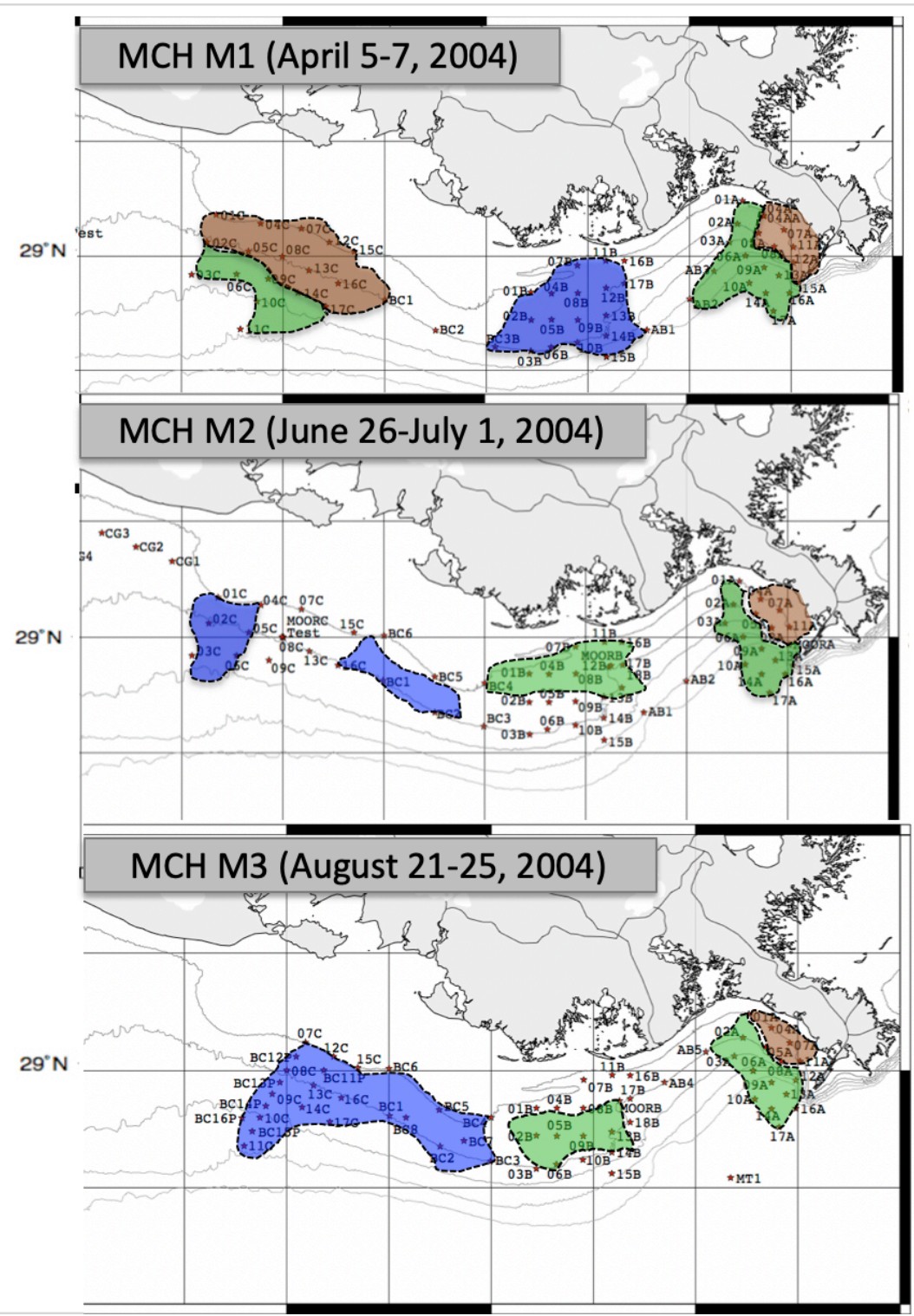

**Figure 8.** Distribution of the three zones during cruises MCH M1-M3 based on salinity data
(Lahiry, 2007). Areas shaded in three colors represent the brown, green and blue zones
respectively.

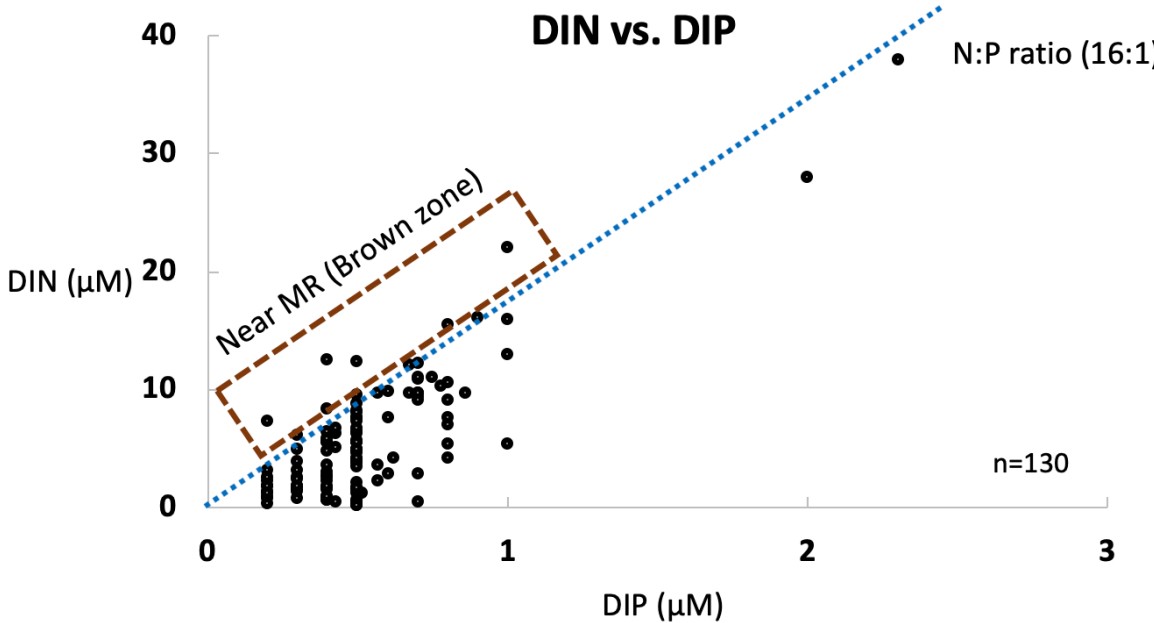

**Figure 9.** Dissolved inorganic nitrogen (DIN) against dissolved inorganic phosphorus (DIP)
during sampling periods in the Gulf of Mexico (GOM) and Mid-western Korea (CSK).
Nearly all samples had an N:P ratio of < 16, which indicated potential N-limited condition.
At a few points near the brown zone the ratio was between 16 -18; this is where light-
limitation is expected according to RC02.

**Table 1.**     Sampling dates for data from Gulf of Mexico projects and the coastal sea of Korea.
Winter data are listed for the Gulf of Mexico cruises.

| Study area | Date | Cruise number |
|---|---|---|
| **Gulf of Mexico** MCH | 5 ~ 7 April 2004 | MCH M1 |
| | 26 June ~ 1 July 2004 | MCH M2 |
| | 21 ~ 25 August 2004 | MCH M3 |
| | 23 ~ 27 March 2005 | MCH M4 |
| | 20 ~ 26 May 2005 | MCH M5 |
| | 23 ~ 29 March 2007 | MCH M8 |
| **Korea** CSK | Feb, May, Aug, Nov (2008) | |




 **Table 2.** Atmospheric Nitrogen Deposition (AN-D) in the USA and in the Yellow Sea.

| Watersheds | AN-D (g m$^{-2}$ year$^{-1}$) | References |
|---|---|---|
| Casco Bay, ME | 0.15 | Castro and Driscoll. 2002 |
| Merrimack River, MA | 0.12 ~ 0.4 | Alexander et al. 2001 |
| Long Island Sound, CT | 0.18 | Castro and Driscoll. 2002 |
| Delaware Bay, DE | 0.22 ~ 0.44 | Castro and Driscoll. 2002<br>Goolsby. 2000 |
| Chesapeake Bay | 0.14 ~ 1.74 | Alexander et al. 2001<br>Castro, M. S et al. 2001<br>Castro and Driscoll. 2002<br>Goolsby. 2000 |
| **Gulf of Mexico** | 1 ~ 1.15 | Wade and Sweet. 2008 |
| **Bohai Sea** | 6.42 ~ 14.25 | Shou et al. 2018 |
| **Yellow Sea (China on the west side)** | 1.61 ~ 1.84<br>2.99 ~ 3.28<br>3.81 ~ 9.24 | Zhao et al. 2015<br>Luo et al. 2014<br>Shou et al. 2018 |
| **Yellow Sea (Korea on the east side)** | 1.5 ~ 5.82 | Kim (JY) et al. 2010 |

Table 3. Definitions and values used in N-mass balance model to calculate DIN removal by biological production. (a) Each one quarter degree box; (b) Wade and Sweet 2008 for GOM region; (c) McCarthy et al., 2015 (d) Lee et al., 2012; (e) McCarthy et al., 2015; (f) Qureshi 1995. *$F_{Atmo}^{DIN}$ of CSK region is used as mean values of Asia data in Table 2, which is initially 5 times higher than that of GOM ($1.4 \times 10^5$ mol day$^{-1}$). ** The unit of $F_{Sink}^{DIN}$ was converted to mol day$^{-1}$ from the unit of original data (gN m$^{-2}$ day$^{-1}$) with area of box (0.25 m x 0.25 m) and molar mass of N (14 g mol$^{-1}$). All unit were converted to mol day$^{-1}$ multiplied by area of box (0.25 m x 0.25 m).

| Unit | Definitions | Value |
|---|---|---|
| $A_{Bott}$ (m$^2$) | Area of box | $6.2 \times 10^8$ m$^2$ (a) |
| $C_{Box}^{DIN}$ (μM) | DIN concentration in each area (box) | |
| $V_S$ (m$^3$) | Water volume of box | $A_{Bott}$ X Pycnocline depth |
| $C_{EX}^{DIN}$ (mmol m$^{-3}$) | Different concentration between each box $C_{EX}$= ($C_{on}$ - $C_{off}$) or ($C_{East}$ − $C_{West}$) for DIN | |
| $\lambda_{Mix}$ (day$^{-1}$) | Mixing rate of each box to box (**A reciprocal of the water residence time**) | |
| $F_{River}$ (day$^{-1}$) | River discharge | |
| $F_{River}^{DIN}$ (mol day$^{-1}$) | DIN flux from each river discharge | |
| $F_{Atmo}^{DIN}$ (mol day$^{-1}$) | Diffusive flux from Atmospheric deposition (Bulk N deposition rate x $A_{Bott}$ ($A_{surface\ of\ ocean}$) for DIN | $1.4 \times 10^5$ mol day$^{-1}$ * (b) |
| $F_{Bott}^{DIN}$ (mol day$^{-1}$) | Benthic flux from the bottom sediments (Net DIN release considered regeneration, groundwater inputs, and uptake of NO$_2$/ NO$_3$) | 1.2 mmol N m$^{-2}$ day$^{-1}$ (c) 6.2 mmol N m$^{-2}$ day$^{-1}$ (d) |
| $F_{Export}^{DIN}$ (mol day$^{-1}$) | An advection term which calculated from the current velocity | |
| $F_{Deni}^{DIN}$ (mol day$^{-1}$) | Denitrification in the water column | 2.1 mmol N m$^{-2}$ day$^{-1}$ (e) |
| $F_{Sink}^{DIN}$ (mol day$^{-1}$) | Vertical sinking of DIN flux from sediment trap data | 0.1 ~ 1 gN m$^{-2}$ day$^{-1}$ ** (f) |
| $F_{Removal}^{DIN}$ (day$^{-1}$) | Removal by biological production (Assuming that the other removal factors are negligible above the pycnocline layer) | |

**Table 4.** Simulation results for selected model scenarios based on MCH M1 (5 ~ 7 April 2004).
Biological production is calculated by our N-mass balance model. Oxygen demand is
calculated by Redfield stoichiometry ratio (C: $-O_2$ = 106: 138) (Unit: gC m$^{-2}$ day$^{-1}$).

| | $F_{River}$ | $F_{AN-D}$ | $F_{Bott/SGD}$ | Biological production | Oxygen demand |
|---|---|---|---|---|---|
| **Nominal Value** | 1.4 x 10$^7$ (~98 %) | 1.4 x 10$^5$ (~1 %) | 1.4 x 10$^5$ (~1 %) | Base line | |
| **Scenario 1** | 5.6 x 10$^6$ (~93 %) | 2.8 x 10$^5$ (~5%) | 1.4 x 10$^5$ (~2%) | ~45% decreased | ~58% decreased |
| **Scenario 2** | 9.8 x 10$^6$ (~96 %) | 2.8 x 10$^5$ (~3%) | 1.4 x 10$^5$ (~1%) | ~22% decreased | ~28% decreased |
| **Scenario 3** | 1.7 x 10$^7$ (~97 %) | 2.8 x 10$^5$ (~2%) | 1.4 x 10$^5$ (~1%) | ~17% increased | ~21% increased |

**Table 5.** Simulation results for selected model scenarios based on CSK (February 2008)
data. Biological production is calculated by our N-mass balance model. Oxygen
demand is calculated by the Redfield stoichiometry ratio (C: $-O_2$ = 106: 138) (Unit: gC m$^{-2}$
day$^{-1}$).

| | $F_{River}$ | $F_{AN-D}$ | $F_{Bott/SGD}$ | Biological production | Oxygen demand |
|---|---|---|---|---|---|
| **Nominal Value** | $1.9 \times 10^6$ (~60%) | $6.0 \times 10^5$ (~20%) | $6.0 \times 10^5$ (~20%) | Base line | |
| **Scenario 1** | $7.2 \times 10^5$ (~29%) | $1.2 \times 10^6$ (~47%) | $6.0 \times 10^5$ (~24%) | ~13% decreased | ~16% decreased |
| **Scenario 2** | $1.3 \times 10^6$ (~41%) | $1.2 \times 10^6$ (~39%) | $6.0 \times 10^5$ (~20%) | ~2% decreased | ~2% decreased |
| **Scenario 3** | $2.2 \times 10^6$ (~55%) | $1.2 \times 10^6$ (~30%) | $6.0 \times 10^5$ (~15%) | ~25% increased | ~32% increased |
