# Peer review of "Implications of different nitrogen input sources for potential production and carbon flux estimates in the coastal Gulf of Mexico (GOM) and Korean coastal waters"

_Ocean Science, 2019_

## Referee Comment (RC1) · Anonymous Referee #1 · 15 Jul 2019

General Comments

The paper addresses the implication of nitrogen sources on ecosystem production in the Gulf of Mexico (GOM) and the coastal sea off Korea (CSK) using a mass-balance approach. It is generally well-written, but there are some confusing aspects. One of the aims of the study is to test a hypothesis about the controls on coastal productivity originally laid out by Rowe and Chapman (2002), which divides coastal waters into brown, green and blue zones, based on productivity. Unfortunately, figure 5 also codes stations in different subregions (each of which span these zones) using a different color

scheme.

The study involves considerable data synthesis using datasets from both the GOM and CSK regions. Perhaps the reason for comparing these two regions is simply the availability of data, but the paper does not otherwise suggest why these particular regions were chosen. Is there a compelling reason to compare only Korean coastal waters with the GOM instead of broadening the comparison to include other data-rich regions (e.g. the Baltic or other European regions)? Some further explanation is required.

The authors have chosen a mass-balance approach, and similar approaches have been used in many other studies. Some earlier modelling studies are cited (lines 23-26) but mass-balance approaches have been used successfully in many regions and individual coastal systems to estimate ecosystem metabolism, nutrient and carbon fluxes (e.g. the large literature generated by the LOICZ project to name a single program, as well as detailed mass balance studies of the Chesapeake Bay, the Baltic and other regions by individual research groups over the years). It seems as if the literature cited could reflect more of this earlier and ongoing work.

I found the presentation of the steady-state mass balance approach to be a little weak in that the equations inadequately representing all the terms present in each of the 3 regions being considered (2 layers for each of the red, brown, and blue sub-compartments). The equations are not well-linked to the figures illustrating processes and transport (figs 2 and 3). The only advective transport terms appears to be that associated with riverine inputs, which presumably occur only in the brown regions, or is this incorrect? Neither layers seem to include advective terms related to upwelling, though upwelling is indicated to be important in some areas (e.g. lines 329-331). In my view, it is better to include all terms specific to each type of compartment rather than to generalize, even if more equations are needed in the text or in supplementary material (perhaps an equation for each layer in each of the blue, green and brown categories? unless they are identical). Also, the compartments appear to be treated as

two-layer, 1-d longitudinal profiles instead of two 2-d layers (lateral extent in 2 d), i.e. the implication of a grid with an upstream and downstream neighbor in open water, not 4 neighbors, one on each face of the gridcell. Can this be clarified? The units of each term in the mass balance are inconsistent, based on the definitions in table 3 (see below). It seems odd that figure 3 illustrates the biogeochemical and transport processes of regions in the GOM, but there is no analogous figure for the Korean coastal waters. Is such a figure assumed to be redundant?

The four "factors" (i.e. assumptions) necessary to run the model (lines 196-203) include the assumptions of steady state, spatial homogeneity, equivalence of biomass and primary production(!) and neglect of denitrification. These assumptions are a bit breathtaking, and at the very least require additional discussion, clarification (specifically, how is primary production rate calculated from chlorophyll measurements) and justification. The carbon equivalent of chlorophyll represents a carbon pool, not a rate of carbon production, so more is needed to estimate primary production than the chlorophyll:C ratio. The absence of consideration of denitrification, especially in regions of high N enrichment such as the Korean waters discussed here, seems strange. Also, the authors note that primary production of coastal waters is jointly controlled by nitrogen and phosphorus (first paragraph of introduction). It would be useful to have some sense of the N:P ratios of these waters to determine whether the assumption of control of productivity by nitrogen is always reasonable, and when it breaks down.

Specific Comments

The mass-balance approach used here consists of three steady-state equations for DIN removal, i.e. net inorganic N uptake associated with biological production (referred to as potential primary production; it seems like a better choice would be overall net ecosystem production). It appears that eq 1 is meant to represent a generic mass balance, and eqs 2 and 3 represent surface and bottom layers (above and below the pycnocline). Where is denitrification? Is it considered part of the sink related to FDIN-removal? FDINsink is obtained from sediment trap data (see table 3). Does this term

(never explicitly defined in the text, but only in table 3) represent organic N particles, adsorbed DIN on mineral sediment, or both?

Eqs 1-3: components of the equations are presented without units, except in table 3, and there they are inconsistent (see below).

Following the development of eqs 1-3, there is some discussion of water transport in the GOM, as if the equations pertain only to this case and not the Korean waters. I think that perhaps the paper should be structured so that the conditions for each site are discussed in parallel sections, as in the Results sections. Table 2 provides estimates of atmospheric N deposition to various watersheds and water bodies from several references for different periods. It is well known that atmospheric N deposition has been declining over most of the US in recent years, and may be increasing or decreasing in Asia, depending upon the locale and period. The authors point out the difference between the increasing trends of N deposition in Korea and the decreasing or flat trends in the GOM, so it seems important to compare the two regions over the same time period. It seems like a better option (or at least a useful additional comparison) would be to include regional N deposition estimates from a global, gridded database over a common period, even if the values are generated by models. Several options exist to obtain such data, including Lamarque et al. (2013).

Table 3 defines some terms not explicitly defined in the text and provides units for the terms. The units shown are not always dimensionally consistent. For example, Friver is given units of 1/days, CDINbox has units of $\mu$m (1e-6 moles/l = 1e-3 moles/m3), and area has units of m2, so that the N flux associated with river input is 1e-3 moles/m/day. FDin removal has units of 1/day, which is inconsistent with this term and that of FDin atmo, which has units of mol/day. The values provided in column 3 of the table are not always in the same units as those in column 1. A good start at fixing this would be to define the units of FDin removal, preferably in the text, and ensure that each of the terms in the equation match the units of the overall equation. Vs, the water volume of a "box" id stated to be the product of bottom area and pycnocline depth. What about

the volume of the bottom layer in eq 3?

The comparison made between figures 6a and 8 in lines 367-373 (Lahiry's salinity-based classification vs that estimated here) is qualitative and unhelpful. Why not indicate the proportion of stations with the same classification in each period, i.e. a quantitative comparison? Rather than straining to say that there is some agreement, why not point out that there really isn't much and why. The salinity-based estimate of a large brown zone in the west in April 2004 is absent in the current estimate, and the large blue region in the center is much smaller. Why should there be much agreement given the differences in the approaches, except where the dominant driver is the massive flow of the Mississippi, which affects both salinity and nutrients? More discussion of this is warranted.

The differences between the above- and below-pycnocline layers are quite evident in the GOM (fig 6a,b) but not so much in the CSK. Specifically, around 90% of the grid cells in figs 7a and 7b (above and below the pycnocline) show the same classification (blue, green, brown) across all months evaluated, but less than half of the grid cells are in agreement in Fig 6a, b. Does this suggest differences in stratification in the GOM and CSK that control the homogeneity of the water column, or other factors? Again, more discussion is warranted.

Technical issues/typos/language

Line 4: phosphorus is misspelled

Bierman et al 1994 is cited a few times in the text, but only Bierman et al 2004 appears in the references. Incorrect year?

Table 2 cites Castro and Driscoll 2002 and Castro M.S. et al. 2000. The references include a Castro et al. 2002 only.

Line 227: The PPP rate wasn't defined….it is the brown zone boundary that is defined as being the region in which PPP is over the 2 g C/m2/day level…at least, this is my

understanding. The text should be modified accordingly.

References cited in this review

Lamarque, J.-F., Dentener, F., McConnell, J., Ro, C.-U., Shaw, M., Vet, R., Bergmann, D., Cameron-Smith, P., Doherty, R., Faluvegi, G., Ghan, S.J., Josse, B., Lee, Y.H., MacKenzie, I.A., Plummer, D., Shindell, D.T., Stevenson, D.S., Strode, S., Zeng, G., 2013. Multi-model mean nitrogen and sulfur deposition from the Atmospheric Chemistry and Climate Model Intercomparison Project (ACCMIP): evaluation historical and projected changes. Atmos. Chem. Phys. 13, 7997–8018. http://dx.doi.org/10.5194/acp-13-7997-2013.
* * *

---

## Referee Comment (RC2) · Anonymous Referee #2 · 7 Aug 2019

General The authors use a simple box model to estimate the potential primary production and associated oxygen demand in to region, the Gulf of Mexico and Korean coastal waters. The aim and scope of the paper seem interesting and valuable for the scientific community. The method, a box model driven by observational data, is generally valid and has been used in numerous publications before. However, the authors apply several simplifying assumptions that could flaw the study: 1) DIN removal equals potential primary production – the ratio of this assumption should at least be explained and critically discussed, 2) assumption of absence of denitrification (even though at

least in GOM there is large hypoxia reported). Thirdly, the hydrodynamical background of mixing between the different compartments/boxes has not been made clear. The authors should think of taking into account modelling works or extensive measuring campaigns presented in the literature in order to deliver a decent foundation for their numbers.

Details - P. 10, ll. 150-152: How are 'output terms for water mixing' calculated in detail? Table 3 says $\lambda$Mix equals the 'reciprocal of residence time'. Is this a realistic approach? In a tidal environment the work done by 'mixing' (dispersion would be more appropriate) increases with residence time instead of being a reciprocal. Maybe this does not apply to GOM and Korean waters but this must be described in detail (dependence of horizontal mixing, i.e. dispersion, on river run-off). - P. 10-11, ll. 160-161: How can gradient of N-concentration between boxes affect the exchange rate? N cannot drive a flow (affect equation of state). - P. 17, ll. 317-318: Why different threshold for 'brown zone' in case of GOM and CSK? "We defined" should result in one definition applying to both regions, otherwise zones can be adjusted by tuning thresholds to give geographically sound 'results' for each region. - P. 18, l. 333: MCK or CSK, what is the correct abbreviation? - Figure 1: Please increase font size of axis tick labels, use approximately same size for all panels - Eq. 1-3: Please include units. - Figure 3: The conditions of "Export N (Mixing)" need some fundamental discussion in the text - Figure 4: This figure is difficult to read. Authors should think of a way to show spatial and dynamical information in one figure; for example they could show a map of GOM (like Fig. 6) with a polar graph representing current speed and direction during one season. The current figure does not really help understand what is happening in time and space. - Table 4: How can EPP be higher than PPP? What does this mean? - Please check citation "Rowe and Capman (2002)". Authors seem to cite wrong title which should read "Continental Shelf Hypoxia: Some nagging questions".

---

## Referee Comment (RC3) · Anonymous Referee #3 · 11 Aug 2019

The manuscript applied a mass balance model based on N to two regions (GOM and CSK) with different N input source to estimate potential primary production (PPP) rate. Although similar box model has been used in many other studies, I still have some concerns about the model in this manuscript: During the peak season of nutrient loading (May to July) of Mississippi-Atchafalaya River, P-limited primary production has been observed in the river plume. In the manuscript, one assumption for the box model is that "DIN is fully utilized by phytoplankton growth". Is this assumption appropriate for the brown zone in GOM? Will the model overestimate biomass in the brown zone in

[Figure]

GOM? Besides the nutrient and light, the temperature is another limitation for phytoplankton growth. In February, climatology SST in the CSK region is around 5 degrees celsius. As shown in Fig 7a, the brown zone has the highest PPP rate in February. I doubt very much if the model suitable for CSK region.

The manuscript declared the current velocity data for the advective flow factor calculation in GOM but didn't for CSK.

The manuscript says that AD-N added N to the surface and enlarged green zone in the CSK all season. The AD-N "mainly came from" the China side. As we knew, this region is under the control of the East-Asian monsoon. If the AD-N primarily came from the China side, it should have a significant seasonal variation because of wind direction changes. Dose the AD-N input in CSK time vary? In Table 2, the authors list AN-D values from references but didn't contain AD-N they used in the model.

In conclusion, "Our results agree well . . . and ocean color remote sensing in the MCK (Son et al., 2005).". Can authors add some details about the comparison in the "results" part?

---

## Author Comment (AC1) · 11 Oct 2019

Thank you for your review and comments. Our responses were uploaded in the attached files. We also highlighted in yellow the changed part in the revised manuscript.

Please also note the supplement to this comment:
https://www.ocean-sci-discuss.net/os-2019-46/os-2019-46-AC1-supplement.zip

---

## Author Response (AR1)

We thank reviewers for their comments on this manuscript. We have tried to address all comments, and please find our responses to comments below.

**Anonymous Referee #1 comments:**

**General Comments**
1.  The paper addresses the implication of nitrogen sources on ecosystem production in the Gulf of Mexico (GOM) and the coastal sea off Korea (CSK) using a mass-balance approach. It is generally well-written, but there are some confusing aspects. One of the aims of the study is to test a hypothesis about the controls on coastal productivity originally laid out by Rowe and Chapman (2002), which divides coastal waters into brown, green and blue zones, based on productivity. Unfortunately, figure 5 also codes stations in different subregions (each of which span these zones) using a different color scheme.

    **Response:** The brown-green-blue hypothesis of Rowe and Chapman is based on distance from the riverine input, as shown in Fig. 5, but for operational reasons stations were occupied in three different regions along the coast, near the Mississippi (red dots, A), an intermediate region (grey dots, B), and near the Atchafalaya (blue dots, C). To prevent a confusion, we added further explanation in the main text (lines 283-285) and caption of Fig. 5 (line 968-970), as follows:

    "For operational and modeling purposes, stations were grouped into three regions – near the Mississippi (A), near the Atchafalaya (C) and an intermediate region between ~90°-91°W."

    Added to caption of Fig. 5: "Red, grey and blue stations correspond to sub-regions A (near the Mississippi River), B (between the Mississippi and Atchafalaya), and C (near the Atchafalaya) respectively."

2.  The study involves considerable data synthesis using datasets from both the GOM and CSK regions. Perhaps the reason for comparing these two regions is simply the availability of data, but the paper does not otherwise suggest why these particular regions were chosen. Is there a compelling reason to compare only Korean coastal waters with the GOM instead of broadening the comparison to include other data-rich regions (e.g. the Baltic or other European regions)? Some further explanation is required.

    **Response:** Thanks for the comments. We were not trying to cover the globe, and the available data were from the CSK and GOM. The main reason to compare the CSK with the GOM because of the difference in nitrogen supply via AN-D. The AN-D source is considerably larger in the CSK region than in the GOM (Wade and Sweet, 2008; Zhao et al., 2015). Comparing our results with data from other regions could be a future effort. We added further explanation in the main text (lines 22-25) as follows:

    "The GOM and CSK were selected in this study because while the major input source to the coastal ocean in both regions is riverine, the AN-D and SGD are considerably more important in the CSK region (Wade and Sweet, 2008; Zhao et al., 2015)."

3. The authors have chosen a mass-balance approach, and similar approaches have been used in many other studies. Some earlier modelling studies are cited (lines 23-26) but mass-balance approaches have been used successfully in many regions and individual coastal systems to estimate ecosystem metabolism, nutrient and carbon fluxes (e.g. the large literature generated by the LOICZ project to name a single program, as well as detailed mass balance studies of the Chesapeake Bay, the Baltic and other regions by individual research groups over the years). It seems as if the literature cited could reflect more of this earlier and ongoing work.

**Response:** Thank you for your suggestions. As we mentioned above response (#2), we focused on the CSK and GOM regions in this paper, not on a global synthesis. Thus, we focused on papers that discuss the hypoxia and physical processes in the GOM and CSK (rather than those that cover estuary and riverine systems). We have added references to some mass balance model studies on other regions as you suggested in the main text (lines 31-37) as follows:

"…and such models have been successfully used in many regions and individual coastal systems to estimate ecosystem metabolism, e.g., in the Patuxent River estuary of the Chesapeake Bay (Hagy et al. 2000; Testa et al., 2008) and in the LOICZ (Land Ocean Interactions in the Coastal Zone) project (e.g., Ramesh et al., 2015). However, there are few such model studies in the GOM and CSK. All previous models for the GOM and the CSK have considered only riverine N as the predominant input source, and no one has considered AN-D as an input in either region."

4. I found the presentation of the steady-state mass balance approach to be a little weak in that the equations inadequately representing all the terms present in each of the 3 regions being considered (2 layers for each of the red, brown, and blue subcompartments).

**Response:** Our original Fig. 3a only showed each input/output terms, so we modified Fig. 3a more correctly based on reviewer's comments (each term contains several factors such as denitrification, vertical sinking etc.) as follow:

[Figure]

Especially, both regions have the same input/output terms although the percentages of each term are different, and both regions include sedimentary denitrification as an output factor in the sub-pycnocline layer box. This factor is included within $F_{Bott}^{DIN}$, which defined as net DIN release from the bottom sediment including nutrient regeneration, groundwater nutrient inputs, and an uptake of nitrate and nitrite by sedimentary denitrification. Also, denitrification from the water column in the bottom box is another significant N removal process, so we used direct measurement of the water column denitrification from the McCarthy et al., (2015) as another output term ($F_{Deni}^{DIN}$) for the GOM. In contrast, in the CSK, we could estimate that there is a very little water column denitrification based on the data of oxygen concentration. Thus, we only considered the sedimentary denitrification factor below the pycnocline layer in the CSK. We explain this more in responses #8 and #9. We added further explanation of each terms correctly in the main text (lines 153-277) with yellow highlights as follow:

[revised manuscript text omitted]

5.  The equations are not well-linked to the figures illustrating processes and transport (figs 2 and 3). The only advective transport terms appears to be that associated with riverine inputs, which presumably occur only in the brown regions, or is this incorrect?

    **Response:** Figure 2 is taken from RC02 and shows the hypothesized physical and biochemical processes that initiate and sustain hypoxia on the Texas-Louisiana Shelf (Rowe and Chapman, 2002). Equations 1-3 are linked with figure 3a and b (GOM) and were applied in the same way to the CSK data. All four faces of each box have input/output advective terms, while the top and bottom of the upper layer include air-sea deposition and a sinking term into the bottom layer respectively. The latter is an input term for the lower layer, while sediment/water exchanges are considered across the bottom face of the lower layer. Thus, riverine input applies only to the inshore boxes in the brown zone. The term Export N (Mixing) incorporates the advective transport term between boxes in 2 dimensions. The values of advection term ($F_{Export}^{DIN}$) in the GOM and CSK showed similar range in the previous studies of each regions (Jacob et al., 2000; Lim et al., 2008; Nowlin et al., 1998a, b), so we applied the same value to both regions. We explained more details of advection term in the main text (lines 166-170, lines 261-266) and the caption of Fig. 3 (lines 954-960) as follows:

    "As an output term, $F_{Export}^{DIN}$ as an advection term was calculated from the current velocity in each region from observations (Nowlin et al., 1998a, b) and from literature data (Jacob et al., 2000; Lim et al., 2008) and the exchange between boxes from the residence time in each box. Note that water and nutrient exchange can take place through all four sides of each box, so the array is two-dimensional."

    "The annual current velocities in the CSK are more affected by tidal exchange and the presence of the Yellow Sea Current, but velocities are similar to those in the GOM (Jacob et al., 2000; Lim et al., 2008). The annual range of the currents is around 0 to 28 cm s$^{-1}$ and 0 to 7 cm s$^{-1}$ for the cross-shelf component. Thus, we used the mean value of the current velocity for the time of year during each cruise in both the GOM and the CSK for calculating the advective flow in both alongshore and onshore/offshore directions."

    Added to caption of Fig. 3: "Export N (Mixing) represents the advective transport term. The processes of biogeochemical and transport processes of both regions are the same and each in/out put factor is the same in the GOM and CSK. Note that transfer between boxes occurs in both directions alongshore and onshore/offshore and is not a one-dimensional process as suggested in the diagram."

6.  Neither layers seem to include advective terms related to upwelling, though upwelling is indicated to be important in some areas (e.g. lines 329-331). In my view, it is better to include all terms specific to each type of compartment rather than to generalize, even if more equations are needed in the text or in supplementary material (perhaps an equation for each layer in each of the blue, green and brown categories? unless they are identical). Also, the compartments appear to be treated as two-layer, 1-d longitudinal profiles instead of two 2-d layers (lateral extent in 2 d), i.e. the implication of a grid with an upstream and downstream neighbor in open water, not 4 neighbors, one on each face of the gridcell. Can this be clarified? The units of each term in the mass balance are inconsistent, based on the definitions in table 3 (see below).

**Response:** We think this is a good suggestion for the future model improvement. However, given that both regions are very shallow and are generally subject to well-established pycnoclines, this suggests that in summer at least, upwelling is not very important. This agrees with a previous study in the GOM (Feng et al., 2014). We added this further explanation in the main text (lines 190-194) as follows:

"We also considered vertical sinking as an input for the sub-pycnocline layer box and as an output from the upper layer. Other possible input factors might be upwelling/downwelling processes; however, these factors are neglected in the model because both regions are shallow and close inshore (Feng et al., 2014; Lim et al., 2008) and we have no observational data on upwelling/downwelling rates."

7. It seems odd that figure 3 illustrates the biogeochemical and transport processes of regions in the GOM, but there is no analogous figure for the Korean coastal waters. Is such a figure assumed to be redundant?

**Response:** Yes, we have assumed that biogeochemical and transport processes of both regions are the same, so we only illustrated the GOM region in Fig. 3. However, the importance of each term is different in the two regions. For example, AN-D occurs in both regions, but the percentage in the GOM is less than that in the CSK region (Wade and Sweet, 2008; Zhao et al., 2015). As we mentioned above response (#5), the values of advection term ($F_{Export}^{DIN}$) in the GOM and CSK showed similar range in the previous studies of each regions (Jacob et al., 2000; Lim et al., 2008; Nowlin et al., 1998a, b), so we applied the same value to both regions. We added this further explanation in the main text (lines 166-170, lines 261-266) and the caption of Fig. 3 (lines 954-960). Please see above response (#5).

8. The four "factors" (i.e. assumptions) necessary to run the model (lines 196-203) include the assumptions of steady state, spatial homogeneity, equivalence of biomass and primary production(!) and neglect of denitrification. These assumptions are a bit breathtaking, and at the very least require additional discussion, clarification (specifically, how is primary production rate calculated from chlorophyll measurements) and justification. The carbon equivalent of chlorophyll represents a carbon pool, not a rate of carbon production, so more is needed to estimate primary production than the chlorophyll:C ratio.

**Response:** We have removed all references to EPP (calculated from the carbon:chlorophyll ratio) from the text and instead now compare our results with published [14]C productivity measurements and satellite-derived estimates of productivity. Actually, we considered sedimentary denitrification within the benthic flux term ($F_{Bott}^{DIN}$) and DIN concentration includes ammonium ($NH_4$), nitrate ($NO_2$), and nitrite ($NO_3$) in our model. Sedimentary denitrification in the lower layer and dissimilatory nitrate reduction to ammonium (DNRA) is an important nitrogen removal process, but regeneration processes such as remineralization provides excess ammonium to overlying water (Lehrter et al., 2012; McCarthy et al., 2015;

Rowe et al., 2002).  Due to this, net DIN flux was calculated and used as the value of $F_{Bott}^{DIN}$ in the GOM and CSK, respectively.  In our model, sedimentary denitrification is not shown as an output term in the equation since it is already considered inside the benthic flux factor.  Denitrification from the water column in the bottom box is also another significant N removal process.  Due to this, we used direct measurement of the water column denitrification from the McCarthy et al., (2015) as another output term ($F_{Deni}^{DIN}$) and added in the Equation 3 for the GOM.  In contrast, in the CSK, we could estimate that there is a very little water column denitrification based on the data of oxygen concentration.  Thus, we only considered the sedimentary denitrification factor below the pycnocline layer in the CSK.  We added further explanation and calculation in the main text (lines 213-236, lines 238-250) as follows:

"In the GOM, benthic sediments provide excess ammonium to overlying water by regeneration processes such as remineralization (Lehrter et al., 2012; Nunnally et al., 2014; Rowe et al., 2002).  Generally, there is an uptake of nitrate and nitrite mainly by sedimentary denitrification (McCarthy et al., 2015) or dissimilatory nitrate reduction to ammonium (DNRA) and assimilation by benthic microalgae (Christensen et al., 2000; Dalsgaard, 2003; Thornton et al., 2007).  Due to this, net DIN flux was used as the value of $F_{Bott}^{DIN}$, which shows DIN release from bottom sediments to overlying water column.  For example, in the GOM, the sum of nitrate and nitrite fluxes to bottom sediments (e.g., May: -10.05, July -61.9, August: -48.42 µmol N m$^{-2}$ h$^{-1}$) were similar or smaller than the flux of ammonium from bottom sediments (e.g., May: 203, July: 152, August: 156 µmol N m$^{-2}$ h$^{-1}$) off Terrebonne bay (McCarthy et al., 2015).  In the CSK, the sum of nitrate and nitrite flux to bottom sediments and ammonium flux are 0.5 ~ 1.4 mmol N m$^{-2}$ d$^{-1}$ and 1.3 ~ 9.6 mmol N m$^{-2}$ d$^{-1}$, respectively, which indicated that excess ammonium with additional nitrate and nitrite were released from sediments in this region (Lee et al., 2012).  The release of nitrate and nitrite in the CSK unlike the GOM can be estimated due to high inputs of nitrogen by groundwater in the CSK (Kim (G) et al., 2011) even though there is minor uptake of nitrate and nitrite.  Diffusion from groundwater can probably be ignored in the GOM as Rabalais et al. (2002) reported that the groundwater discharge is very low in coastal Louisiana, but is likely important elsewhere and is known to be important in the CSK.  Based on this, we averaged and sum the fluxes data of nitrate, nitrite, and ammonium from McCarthy et al., 2015 for the GOM and Lee et al., 2012 for the CSK, respectively, and then applied $F_{Bott}^{DIN}$ value as 1.2 mmol N m$^{-2}$ day$^{-1}$ in the GOM and 6.2 mmol N m$^{-2}$ day$^{-1}$ in the CSK.  Thus, in equation 3, the benthic flux term is calculated from existing literature results after considering all DIN fluxes as above (Lee et al., 2012; McCarthy et al., 2015), and then multiplied by the area of each box."

"2) $F_{Deni}^{DIN}$, the denitrification rate from the water column.  Due to high stratification at the pycnocline, upward transfer of dissolved material from the lower layer to the upper layer is assumed not to occur in our model.  Also, denitrification from the water column below the pycnocline is a significant N removal process, which removes up to a maximum 68% of total N input from the Mississippi River in the GOM (McCarthy et al., 2015).  As the value of $F_{Deni}^{DIN}$ in the GOM, we used a direct measurement of denitrification rates from the McCarthy et al., (2015) in the water column (88 µmol m$^{-2}$ h$^{-1}$, which converted to 2.1 mmol N m$^{-2}$ day$^{-1}$) where the stations were exactly same as our sub-region A, B, and C.  We assumed this applied only below the pycnocline where oxygen concentrations decrease.  However, in the CSK, there is no water column denitrification data because the dissolved oxygen concentration has never been down below about 4 mg L$^{-1}$ during our data periods.  Based on this, we estimated that there is a very little water column denitrification in the CSK, so we did not count this term in the CSK.  Thus, we only considered the sedimentary denitrification term for the CSK region."

Even with this assumption, our model result (PPP) is similar to measured rates from $^{14}$C incubation reported by Quigg et al., 2011 in the three subregions A, B, and C.  In the future study, we will fully consider incubation comparison in the same station.  However, at this point, our assumption and all we can do are provided and our model results based on the assumption are similar range with Actual primary production from $^{14}$C incubation data from previous literature paper (Dagg et al., 2007; Lohrenz et al., 1998, 1999; Redalje et al., 1994).  We added further explanation in the main text (lines 333-349, 499-511) as follows:

 "Additionally, Quigg et al., (2011) pointed out that these higher PP values were affected by high riverine nutrients input from the MR that flows westward during that time period.
    The actual PP ranges were similar with our model-based PPP (Figure 6).  However, this was different from RC02's brown zone.  This might be due to the differences between methods such as $^{14}$C, our N-mass balance model, and RC02's theoretical model.  Typically, RC02 assumed that the brown zone is light limited due to high sediment turbidity, but our model does not account for this and only considered DIN concentrations.  Except for this, our PPP results are similar to direct productivity measurements from the $^{14}$C incubations (Quigg et al., 2011).  Our model result (PPP) showed the same range of values as $^{14}$C incubations (e.g., Dagg et al., 2007; Lohrenz et al., 1998, 1999; Quigg et al., 2011; Redalje et al., 1994) in the three sub-regions.
    Note that our model assumed all the biological uptake could be converted directly to production rates, which we considered as PPP.  The PPP from cruises MCH M1 ~ M8 for samples from above the pycnocline calculated using our model is reasonable based on comparison with previous PP values (Figure 6a).  The PPP ranges (0.01 ~ 5.05 gC m$^{-2}$ day$^{-1}$) were similar to previous $^{14}$C measurement PP values of between 0.04 ~ 5.9 gC m$^{-2}$ day$^{-1}$."

"RC02 considered their model to be theoretical.  In the brown zone, close to the river mouth, they assumed turbidity leads to light-limited conditions.  Their results agree well with measured $^{14}$C PP numbers from Quigg et al. (2011) who found the lowest integrated PP is near the MR delta mouth.  However, our N mass balance model did not consider light limitation and therefore PPP in the brown zone is high.  Such good agreement suggests that our model can be applied to a wide region, while $^{14}$C measurements are typically conducted at a few specific points, as long as such limitations are taken into account.
    In the CSK, most previous production studies focused on inshore areas such as estuaries or rivers.  Our research focused for the first time on the coastal ocean off Korea.  Our results explained that diverse nitrogen sources need to be recognized as potential issues for future nutrient management concerned with hypoxia, eutrophication, or other environmental issues.  The agreement between our results and the pattern of production based on satellite-sensing in the CSK (Son et al., 2005), suggests that our model is reasonable."

9. The absence of consideration of denitrification, especially in regions of high N enrichment such as the Korean waters discussed here, seems strange. Also, the authors note that primary production of coastal waters is jointly controlled by nitrogen and phosphorus (first paragraph of introduction). It would be useful to have some sense of the N:P ratios of these waters to determine whether the assumption of control of productivity by nitrogen is always reasonable, and when it breaks down.

**Response:** As we mentioned above response (#8), sedimentary denitrification was considered within a net DIN flux term ($F_{Bott}^{DIN}$) and water column denitrification was added as another output term ($F_{Deni}^{DIN}$) in our model. We added further explanation in the main text (lines 214-237, lines 239-254). Please see our answer to response (#8).

We have also added something on using N:P ratios to determine nutrient limitation. As we explain, in the coastal GOM especially near the Mississippi delta mouth, the P may be more important than N (Sylvan et al., 2006) depending on the time of year (lines 10-11). However, previous studies demonstrated that both GOM and CSK regions are N-limited most of the time (Kim (G) et al., 2011; Turner and Rabalais, 2013). Related to this, we confirmed N-limited condition in our study regions in the GOM and CSK based on plotting N against P. Minimum P concentrations were about 0.5 µM/L which suggests there was always enough P for continued phytoplankton growth. We added a new figure (Fig. 9) and further explanation in the main text (lines 471-480) as follows:

[Figure]

**Figure 9.** DIN against DIP during sampling periods in the GOM and CSK. Nearly all samples had an N:P ratio of < 16, which indicated potential N-limited condition. At a few points near the brown zone the ratio was between 16 -18; this is where light-limitation is expected according to RC02.

"Our results also agree with previous studies that demonstrated that both the GOM and CSK regions are N-limited for most of the year (Kim (G) et al., 2011; Turner and Rabalais, 2013). This compares with the results of Sylvan et al., (2007), who reported that the coastal GOM could be P-limited in the MR delta mouth area where our brown zone is located, while RC02 suggested light-limitation rather than N- or P-limitation. However, this P-limited condition appears to occur when N concentrations are very high. In particular, the N/P ratios in the both the GOM and CSK during our sampling were less than 16, indicating that both regions were N-limited, although a few stations in the brown zone near the MR river area had ratios of between 16 and 18 (Figure 9). These higher N/P ratios may result from the high sediment turbidity causing light-limited conditions in this zone near the river mouth (Rowe and Chapman, 2002)."

**Specific Comments**

1. The mass-balance approach used here consists of three steady-state equations for DIN removal, i.e. net inorganic N uptake associated with biological production (referred to as potential primary production; it seems like a better choice would be overall net ecosystem production). It appears that eq 1 is meant to represent a generic mass balance, and eqs 2 and 3 represent surface and bottom layers (above and below the pycnocline). Where is denitrification? Is it considered part of the sink related to FDINremoval?

   **Response:** Yes, Equation 1 is total consideration for upper/lower boxes and Equations 2 and 3 are applicable to the surface and bottom layers respectively. $F_{Sink}^{DIN}$ is considered to be an output term from the upper layer and an input term for the lower layer. In the lower layer this includes contributions from both sinking and sediment-water transfer. For denitrification factor, we already considered as another output term below the pycnocline layer as we explained above. Please see our responses to #8 and #9. We added further explanation related to our assumption in the main text (lines 270-272) as follows:

   "…in the layer below the pycnocline, as we mentioned above, denitrification, which leads to a main loss of DIN as nitrogen gas, is considered as another output term in Equation 3."

2. FDINsink is obtained from sediment trap data (see table 3). Does this term (never explicitly defined in the text, but only in table 3) represent organic N particles, adsorbed DIN on mineral sediment, or both?

   **Response:** Yes, vertical sinking data is from Qureshi (1995). Based on her dissertation, the number represents Organic N.

3. Eqs 1-3: components of the equations are presented without units, except in table 3, and there they are inconsistent (see below #5).

   **Response:** When we run our model calculation, we made sure to consider the same units of factors. We have added units for all factors in Table 3. For example, to calculate the total equation 1-3, we converted the unit of $F_{Sink}^{DIN}$ term to mol/day from gN m$^{-2}$ day$^{-1}$(original data from Qureshi (1995) with area of box (0.25 x 0.25 size) and mol-N from gN using molar mass of N (14 g/mol). We added details in the caption of Table 3 (lines 1004-1007) as follows:

   Added to caption of Table 3: "** The unit of $F_{Sink}^{DIN}$ was converted to mol day$^{-1}$ from the unit of original data (gN m$^{-2}$ day$^{-1}$) with area of box (0.25 m x 0.25 m) and molar mass of N (14 g mol $^{-1}$). All unit were converted to mol day$^{-1}$ multiplied by area of box (0.25 m x 0.25 m)."

4. Following the development of eqs 1-3, there is some discussion of water transport in the GOM, as if the equations pertain only to this case and not the Korean waters. I think that perhaps the paper should be structured so that the conditions for each site are discussed in parallel sections, as in the Results sections. Table 2 provides estimates of atmospheric N deposition to various watersheds and water bodies from several references for different periods. It is well known that atmospheric N deposition has been declining over most of the US in recent years, and may be increasing or decreasing in Asia, depending upon the locale and period. The authors point out the difference between the increasing trends of N deposition in Korea and the decreasing or flat trends in the GOM, so it seems important to compare the two regions over the same time period. It seems like a better option (or at least a useful additional comparison) would be to include regional N deposition estimates from a global, gridded database over a common period, even if the values are generated by models. Several options exist to obtain such data, including Lamarque et al. (2013).

   **Response:** Please see above response (#7). We added this further explanation in the main text (main text (lines 166-170, lines 261-266) and the caption of Fig. 3 (lines 954-960). Please see above response (#5).

   Our model calculation is based on the observational data, not model simulation. Comparing our results with model simulation from previous studies could be a future effort. So, for the AN-D concentration, we have AN-D measurement data (observational data) from Sweet and Wade 2008 in the GOM and from Kim (JY) et al., 2010 in Korea. The important thing is that the AN-D concentrations are quite different in both regions (Table 2). Thus, we used different value of $F_{Atmo}^{DIN}$ for the GOM and CSK and added further explanation in the main text (lines 187-194) and the caption of Table 3 (line 1003-1004) as follows:

   "The mean value of Asian data, as shown in Table 2 (Kim (JY) et al., 2010; Luo et al., 2014; Shou et al., 2018; Zhao et al., 2015), is used for $\boldsymbol{F_{Atmo}^{DIN}}$ of the CSK region, which is initially five times higher than that of the GOM (1.4 X $10^5$ mol day$^{-1}$; Wade and Sweet, 2008). We also considered vertical sinking as an input for the sub-pycnocline layer box and as an output from the upper layer. Other possible input factors might be upwelling/downwelling processes; however, these factors are neglected in the model because both regions are shallow and close inshore (Feng et al., 2014; Lim et al., 2008) and we have no observational data on upwelling/downwelling rates."

   Added to caption of Table. 3: "*$\boldsymbol{F_{Atmo}^{DIN}}$ of CSK region is used as mean values of Asia data in Table 2, which is initially 5 times higher than that of GOM (1.4 X $10^5$ mol day$^{-1}$)."

5. Table 3 defines some terms not explicitly defined in the text and provides units for the terms. The units shown are not always dimensionally consistent. For example, Friver is given units of 1/days, CDINbox has units of m (1e-6 moles/l = 1e-3 moles/m3), and area has units of m2, so that the N flux associated with river input is 1e-3 moles/m/day. FDin removal has units of 1/day, which is inconsistent with this term and that of FDin atmo, which has units of mol/day. The values provided in column 3 of the table are not always in the same units as those in column 1. A good start at fixing this would be to define the units of FDin removal, preferably in the text, and ensure that each of the terms in the equation match the units of the overall equation. Vs, the water volume of a "box" id stated to be the product of bottom area and pycnocline depth. What about the volume of the bottom layer in eq 3?

**Response:** As we mentioned above response (#3 of specific comments), we converted the unit of $F_{Sink}^{DIN}$ to mol day$^{-1}$ from the original data 0.1 ~ 1 gN m$^{-2}$ day$^{-1}$. We added this explanation in the caption of Table 3 (lines 1004-1005). The volume of the bottom layer in equation 3 is area multiplied by each sampling data's pycnocline depth.

6. The comparison made between figures 6a and 8 in lines 367-373 (Lahiry's salinity-based classification vs that estimated here) is qualitative and unhelpful. Why not indicate the proportion of stations with the same classification in each period, i.e. a quantitative comparison? Rather than straining to say that there is some agreement, why not point out that there really isn't much and why. The salinity-based estimate of a large brown zone in the west in April 2004 is absent in the current estimate, and the large blue region in the center is much smaller. Why should there be much agreement given the differences in the approaches, except where the dominant driver is the massive flow of the Mississippi, which affects both salinity and nutrients? More discussion of this is warranted.

**Response:** We would expect better agreement, since given that hypoxia is supposedly dependent on nitrate input, which is related to freshwater flow and stratification, even if the relationship is not simple. We revised the discussion to compare our results with previous literatures more detailed in the main text (lines 456-480) as follows:

"Our zonal boundaries can be compared with the results of Lahiry (2007), who used salinity to define the edges of each zone for the three cruises MCH M1, M2, and M3 (Figure 8) and defined the edges of the RC02 zones in the coastal GOM based solely on salinity. Her limited simulation results indicated similar patterns to our model based on DIN concentration near the Mississippi River mouth (e.g., during MCH M1, M2, and M3). Mixing was more conservative in this region than further west because the low salinity water with high nutrients was less diluted with offshore water.

Away from the MR in sub-regions B and C, however, her results gave very different boundaries for the three zones compared with our results (Figure 8). In particular, the results near the Atchafalaya River were very different (compare Figures 6 and 8). For example, our data showed only green and blue zones off Atchafalaya Bay during MCH M1, with no brown zone. Similarly, the extent of the blue zones in sub-region C during MCH M2 and M3 is also very different. We believe that our DIN-model based classification can cover more complex biological processes than the Lahiry (2007) method, which considers only advection and mixing and the DIN-model is a more sensible way to look at biological processes in the GOM.

Our results also agree with previous studies that demonstrated that both the GOM and CSK regions are N-limited for most of the year (Kim (G) et al., 2011; Turner and Rabalais, 2013). This compares with the results of Sylvan et al., (2007), who reported that the coastal GOM could be P-limited in the MR delta mouth area where our brown zone is located, while RC02 suggested light-limitation rather than N- or P-limitation. However, this P-limited condition appears to occur when N concentrations are very high. In particular, the N/P ratios in the both the GOM and CSK during our sampling were less than 16, indicating that both regions were

N-limited, although a few stations in the brown zone near the MR river area had ratios of between 16 and 18 (Figure 9). These higher N/P ratios may result from the high sediment turbidity causing light-limited conditions in this zone near the river mouth (Rowe and Chapman, 2002)."

7. The differences between the above- and below-pycnocline layers are quite evident in the GOM (fig 6a,b) but not so much in the CSK. Specifically, around 90% of the grid cells in figs 7a and 7b (above and below the pycnocline) show the same classification (blue, green, brown) across all months evaluated, but less than half of the grid cells are in agreement in Fig 6a, b. Does this suggest differences in stratification in the GOM and CSK that control the homogeneity of the water column, or other factors? Again, more discussion is warranted.

**Response:** We believe this is due to different amounts of freshwater flow in both regions. Typically, because the MARS provides a lot more freshwater to the GOM than the local rivers do to the CSK, we can get more stratification in the GOM than off Korea. Our model results in the CSK (Fig. 7a, b) show 90% agreement for the boxes above and below the pycnocline layer. We added further explanation in the main text (lines 400-414) as follow:

"Around 90% of the grid cells in the CSK are in the same zones above and below the pycnocline (Figure 7 a and b) during all four cruises; however, in the GOM (Figure 6 a and b) this was found for fewer than half of the grid cells. This is probably due to the difference in freshwater discharge rate in the two regions, which leads to a much larger stratified area in the GOM than in the CSK.

One question that has not been investigated is the temperature dependence of primary productivity in the two areas. While the GOM is temperate throughout the year, winter temperatures in the CSK fall to ~5°C. However, according to the ocean color remote sensing images from near the CSK river mouth reported by Son et al., (2005), primary production in the CSK does not appear to be strongly affected by temperature. The PPP results of our model ($0.2$ to $2.2$ gC m$^{-2}$ day$^{-1}$) agreed with their ocean color remote sensing results ($0.4$ to $1.6$ gC m$^{-2}$ day$^{-1}$) in the CSK. Also, during all seasons, the Keum River consistently supplies high amounts of DIN (average: $< 60$ $\mu$M) (Lim et al., 2008) to the coastal zone (especially close to the Keum mouth). We believe, therefore, that the higher value of PPP in winter near the Keum mouth (brown zone in figure 7a), is reasonable."

**Technical issues/typos/language**
1. Line 4: phosphorus is misspelled

**Response:** Yes. We corrected the misspelled word (line 4).

2. Bierman et al 1994 is cited a few times in the text, but only Bierman et al 2004 appears in the references. Incorrect year?

**Response:** Yes. We corrected.

3.  Table 2 cites Castro and Driscoll 2002 and Castro M.S. et al. 2000. The references include a Castro et al. 2002 only.

    **Response:** Yes. We corrected.

4.  Line 227: The PPP rate wasn't defined: : :.it is the brown zone boundary that is defined as being the region in which PPP is over the 2 g C/m2/day level: : :at least, this is my understanding. The text should be modified accordingly.

    **Response:** Yes. We modified the text.

    The actual PP ranges were similar with our model-based PPP (Figure 6). However, this was different from RC02's brown zone. This might be due to the differences between methods such as $^{14}$C, our N-mass balance model, and RC02's theoretical model. Typically, RC02 assumed that the brown zone is light limited due to high sediment turbidity, but our model does not account for this and only considered DIN concentrations. Except for this, our PPP results are similar to direct productivity measurements from the $^{14}$C incubations (Quigg et al., 2011). Our model result (PPP) showed the same range of values as $^{14}$C incubations (e.g., Dagg et al., 2007; Lohrenz et al., 1998, 1999; Quigg et al., 2011; Redalje et al., 1994) in the three sub-regions.
    Note that our model assumed all the biological uptake could be converted directly to production rates, which we considered as PPP. The PPP from cruises MCH M1 ~ M8 for samples from above the pycnocline calculated using our model is reasonable based on comparison with previous PP values (Figure 6a). The PPP ranges (0.01 ~ 5.05 gC m$^{-2}$ day$^{-1}$) were similar to previous $^{14}$C measurement PP values of between 0.04 ~ 5.9 gC m$^{-2}$ day$^{-1}$."

"RC02 considered their model to be theoretical. In the brown zone, close to the river mouth, they assumed turbidity leads to light-limited conditions. Their results agree well with measured [14]C PP numbers from Quigg et al. (2011) who found the lowest integrated PP is near the MR delta mouth. However, our N mass balance model did not consider light limitation and therefore PPP in the brown zone is high. Such good agreement suggests that our model can be applied to a wide region, while [14]C measurements are typically conducted at a few specific points, as long as such limitations are taken into account.

In the CSK, most previous production studies focused on inshore areas such as estuaries or rivers. Our research focused for the first time on the coastal ocean off Korea. Our results explained that diverse nitrogen sources need to be recognized as potential issues for future nutrient management concerned with hypoxia, eutrophication, or other environmental issues. The agreement between our results and the pattern of production based on satellite-sensing in the CSK (Son et al., 2005), suggests that our model is reasonable."

**Response: 2)** Sedimentary denitrification was included in the bottom water boxes but was not fully explained in the text. As we mentioned in another response (reviewer 1, #8 of general comments), net DIN flux was calculated including sedimentary denitrification and regeneration process and used as the value of $F_{Bott}^{DIN}$ in the GOM and CSK, respectively. Also, denitrification from the water column in the bottom box is another significant N removal process. Due to this, we used direct measurement of the water column denitrification from the McCarthy et al., (2015) as another output term ($F_{Deni}^{DIN}$) and added in the Equation 3 for the GOM. However, in the CSK, we could estimate that there is a very little water column denitrification based on the data of oxygen concentration. Thus, we only considered the sedimentary denitrification factor below the pycnocline layer in the CSK. We added further explanation and calculation in the main text (lines 213-236, lines 238-250) as follows:

"In the GOM, benthic sediments provide excess ammonium to overlying water by regeneration processes such as remineralization (Lehrter et al., 2012; Nunnally et al., 2014; Rowe et al., 2002). Generally, there is an uptake of nitrate and nitrite mainly by sedimentary denitrification (McCarthy et al., 2015) or dissimilatory nitrate reduction to ammonium (DNRA) and assimilation by benthic microalgae (Christensen et al., 2000; Dalsgaard, 2003; Thornton et al., 2007). Due to this, net DIN flux was used as the value of $F_{Bott}^{DIN}$, which shows DIN release from bottom sediments to overlying water column. For example, in the GOM, the sum of nitrate and nitrite fluxes to bottom sediments (e.g., May: -10.05, July -61.9, August: -48.42 µmol N m$^{-2}$ h$^{-1}$) were similar or smaller than the flux of ammonium from bottom sediments (e.g., May: 203, July: 152, August: 156 µmol N m$^{-2}$ h$^{-1}$) off Terrebonne bay (McCarthy et al., 2015). In the CSK, the sum of nitrate and nitrite flux to bottom sediments and ammonium flux are 0.5 ~ 1.4 mmol N m$^{-2}$ d$^{-1}$ and 1.3 ~ 9.6 mmol N m$^{-2}$ d$^{-1}$, respectively, which indicated that excess ammonium with additional nitrate and nitrite were released from sediments in this region (Lee et al., 2012). The release of nitrate and nitrite in the CSK unlike the GOM can be estimated due to high inputs of nitrogen by groundwater in the CSK (Kim (G) et al., 2011) even though there is minor uptake of nitrate and nitrite. Diffusion from groundwater can probably be ignored in the GOM as Rabalais et al. (2002) reported that the groundwater discharge is very low in coastal Louisiana, but is likely important elsewhere and is known to be important in the CSK. Based on this, we averaged and sum the fluxes data of nitrate, nitrite, and ammonium from McCarthy et al., 2015 for the GOM and Lee et al., 2012 for the CSK, respectively, and then applied $F_{Bott}^{DIN}$ value as 1.2 mmol N m$^{-2}$ day$^{-1}$ in the GOM and 6.2 mmol N m$^{-2}$ day$^{-1}$ in the CSK.  Thus, in equation 3, the benthic flux term is calculated from existing literature results after considering all DIN fluxes as above (Lee et al., 2012; McCarthy et al., 2015), and then multiplied by the area of each box."

"2) $F_{Deni}^{DIN}$, the denitrification rate from the water column.  Due to high stratification at the pycnocline, upward transfer of dissolved material from the lower layer to the upper layer is assumed not to occur in our model.  Also, denitrification from the water column below the pycnocline is a significant N removal process, which removes up to a maximum 68% of total N input from the Mississippi River in the GOM (McCarthy et al., 2015).  As the value of $F_{Deni}^{DIN}$ in the GOM, we used a direct measurement of denitrification rates from the McCarthy et al., (2015) in the water column (88 µmol m$^{-2}$ h$^{-1}$, which converted to 2.1 mmol N m$^{-2}$ day$^{-1}$) where the stations were exactly same as our sub-region A, B, and C.  We assumed this applied only below the pycnocline where oxygen concentrations decrease.  However, in the CSK, there is no water column denitrification data because the dissolved oxygen concentration has never been down below about 4 mg L$^{-1}$ during our data periods.  Based on this, we estimated that there is a very little water column denitrification in the CSK, so we did not count this term in the CSK.  Thus, we only considered the sedimentary denitrification term for the CSK region."

**Response: 3)** This was not explained well in the text, but has now been corrected.  The formulation of the advective terms, which are carried across all four walls of each box to give a 2-D description, rather than the 1-D shown in Fig. 3.  As we mentioned in another response (reviewer 1, #5 of general comments), all four faces of each box have input/output advective terms, while the top and bottom of the upper layer include air-sea deposition and a sinking term into the bottom layer respectively.  The latter is an input term for the lower layer, while sediment/water exchanges are considered across the bottom face of the lower layer.  Thus, riverine input applies only to the inshore boxes in the brown zone.  The term Export N (Mixing) incorporates the advective transport term between boxes in 2 dimensions.  The values of advection term ($F_{Export}^{DIN}$) in the GOM and CSK showed similar range in the previous studies of each regions (Jacob et al., 2000; Lim et al., 2008; Nowlin et al., 1998a, b), so we applied the same value to both regions.  We added this information in the main text (lines 166-170, lines 261-266) and the caption of Fig. 3 (lines 954-960) as follows:

"As an output term, $F_{Export}^{DIN}$ as an advection term was calculated from the current velocity in each region from observations (Nowlin et al., 1998a, b) and from literature data (Jacob et al., 2000; Lim et al., 2008) and the exchange between boxes from the residence time in each box. Note that water and nutrient exchange can take place through all four sides of each box, so the array is two-dimensional."

"The annual current velocities in the CSK are more affected by tidal exchange and the presence of the Yellow Sea Current, but velocities are similar to those in the GOM (Jacob et al., 2000; Lim et al., 2008).  The annual range of the currents is around 0 to 28 cm s$^{-1}$ and 0 to 7 cm s$^{-1}$ for the cross-shelf component.  Thus, we used the mean value of the current velocity for the time of year during each cruise in both the GOM and the CSK for calculating the advective flow in both alongshore and onshore/offshore directions."

Added to caption of Fig. 3: "Export N (Mixing) represents the advective transport term. The processes of biogeochemical and transport processes of both regions are the same and each in/out put factor is the same in the GOM and CSK. Note that transfer between boxes occurs in both directions alongshore and onshore/offshore and is not a one-dimensional process as suggested in the diagram."

**Details**

1. P. 10, ll. 150-152: How are 'output terms for water mixing' calculated in detail? Table 3 says Mix equals the 'reciprocal of residence time'. Is this a realistic approach? In a tidal environment the work done by 'mixing' (dispersion would be more appropriate) increases with residence time instead of being a reciprocal. Maybe this does not apply to GOM and Korean waters but this must be described in detail (dependence of horizontal mixing, i.e. dispersion, on river run-off).

   **Response:** Tidal variability in the GOM is actually small (the tidal range is only ~50 cm along the northern Gulf coast) and tidal mixing in this region is considerably less than that from local currents, which are largely wind-driven (Feng et al., 2012, 2014). Reciprocal residence time has been used previously in models (e.g., De Boer A.M. et al., 2010; Kim (G) et al., 2011). Also, when we calculated the residence time factor, we already fully considered horizontal mixing based on river discharge speed, run-off, dispersion, current velocity etc. We explained more details in the revision. Please see more details in lines 170-174.

   "$F_{Export}^{DIN}$ for water mixing was calculated from these factors; $C_{EX}^{DIN}$ is the difference in DIN concentration between adjacent boxes, $V_S$ is the water volume of each box, and $\lambda_{Mix}$ is the mixing rate of each box $\left( C_{EX}^{DIN} \times V_S \times \lambda_{Mix} \right)$. We used a reciprocal of the water residence time that we considered to represent horizontal mixing, i.e. dispersion."

2. P. 10-11, ll. 160-161: How can gradient of N-concentration between boxes affect the exchange rate? N cannot drive a flow (affect equation of state).

   **Response:** First, we considered the difference of N-Concentrations in each box from our observational data. Then, we checked how much N concentrations were changed between each box to box. To do this, we assumed that during mixing or water flow, the changed concentration of DIN is due to the mixing factor and biological uptake.

3. P. 17, ll. 317-318: Why different threshold for 'brown zone' in case of GOM and CSK? "We defined" should result in one definition applying to both regions, otherwise zones can be adjusted by tuning thresholds to give geographically sound 'results' for each region.

   **Response:** We determined the threshold for each zone from our model results. There is no reason that the brown zone in each region should produce the same threshold value for PPP, since the riverine input (the main source of N in both regions) contains different DIN concentrations, while the river discharge also varies considerably. Our GOM results appear reasonable based on previous studies that defined a boundary between the green and blue zones (Dagg and Breed 2003; Lohrenz et al., 1999).

4. P. 18, l. 333: MCK or CSK, what is the correct abbreviation?

    **Response:** Study area in Korea is part of Mid-western CSK (coastal sea off Korea). We fixed wording CSK instead of MCK. Typically, we only used in Figure 7a, b as MCK due to area is pointed out Mid-western CSK (MCK).

5. Figure 1: Please increase font size of axis tick labels, use approximately same size for all panels.

    **Response:** We will fix in the final revision version.

6. Eq. 1-3: Please include units.

    **Response:** Yes, we put the units of the equations in the Table 3. This point was addressed this point in another response (reviewer 1, #3 of specific comments). We added details in the caption of Table 3 (lines 1004-1006) as follows:

    Added to caption of Table 3: "** The unit of $F_{Sink}^{DIN}$ was converted to mol day$^{-1}$ from the unit of original data (gN m$^{-2}$ day$^{-1}$) with area of box (0.25 m x 0.25 m) and molar mass of N (14 g mol $^{-1}$)."

7. Figure 3: The conditions of "Export N (Mixing)" need some fundamental discussion in the text.

    **Response:** We have modified our original Fig. 3a to be more representative. Please see our response to reviewer 1, #4 of general comments. The term Export N (Mixing) incorporates the advective transport term between boxes in 2 dimensions. As we mentioned in above response (#1 of general, especially response 3)), the values of advection term ($F_{Export}^{DIN}$) in the GOM and CSK showed similar range in the previous studies of each regions (Jacob et al., 2000; Lim et al., 2008; Nowlin et al., 1998a, b), so we applied the same value to both regions. We explained more details of advection term in the main text (lines 166-170, lines 261-266) and the caption of Fig. 3 (lines 954-960).

8. Figure 4: This figure is difficult to read. Authors should think of a way to show spatial and dynamical information in one figure; for example, they could show a map of GOM (like Fig. 6) with a polar graph representing current speed and direction during one season. The current figure does not really help understand what is happening in time and space.

    **Response:** The current data is taken from a large program that was carried out over three years along the Texas-Louisiana shelf. We have shown the mean current speeds and their standard deviations for fortnightly intervals throughout the program. The data show clearly the change from onshore currents moving to the east during summer to generally alongshore to the west during non-summer periods. While we could have shown the data as a series of vector plots, we believe that Fig. 4 is quite comprehensible.

9. Table 4: How can EPP be higher than PPP? What does this mean?

**Response:** We have removed all references to EPP from the manuscript as it does not affect how the model operates and was used only for comparison with model output. Instead, we have compared our model results with spot measurements of primary production using $^{14}$C and with estimates from satellite imagery.

10. Please check citation "Rowe and Chapman (2002)". Authors seem to cite wrong title which should read "Continental Shelf Hypoxia: Some nagging questions"

    **Response:** Yes. I corrected.

   **Response:** Not in our model. We fully agreed that AN-D in the CSK depends on wind direction, but we used the same AN-D concentration in each season because there is not enough observational data from both sides (Korea and China). Thus, at this point, all we want to say is that AN-D contributes considerably in the CSK region and this may need to be considered in future work. Based on this, we used the mean values of Asian data in table 2, which is initially 5 times higher than that of GOM.  As we addressed this point in another response (reviewer 1, #4 of specific comments), we used different value of $F_{Atmo}^{DIN}$ for the GOM and CSK and we added this information in the main text (lines 187-194) and the caption of Table 3 (lines 1003-1004) as follows:

"The mean value of Asian data, as shown in Table 2 (Kim (JY) et al., 2010; Luo et al., 2014; Shou et al., 2018; Zhao et al., 2015), is used for $\mathbf{F_{Atmo}^{DIN}}$ of the CSK region, which is initially five times higher than that of the GOM (1.4 X $10^5$ mol day$^{-1}$; Wade and Sweet, 2008).  We also considered vertical sinking as an input for the sub-pycnocline layer box and as an output from the upper layer.  Other possible input factors might be upwelling/downwelling processes; however, these factors are neglected in the model because both regions are shallow and close inshore (Feng et al., 2014; Lim et al., 2008) and we have no observational data on upwelling/downwelling rates."

Added to caption of Table. 3: "*$\mathbf{F_{Atmo}^{DIN}}$ of CSK region is used as mean values of Asia data in Table 2, which is initially 5 times higher than that of GOM (1.4 X $10^5$ mol day$^{-1}$)."

5.  In conclusion, "Our results agree well : : : and ocean color remote sensing in the MCK (Son et al., 2005)." Can authors add some details about the comparison in the "results" part?

**Response:** This comparison is based on color remote sensing imagery from Son's paper and our predicted model results (son et al., 2010).  As we mentioned above response (#2), the PPP results of our model agreed with their ocean color remote sensing results in this region (Son et al., 2010).  This is described in the discussion part of revision to explain why our model is suitable in the CSK (line 405-414).  Please see above response (#2).

**References**

[revised manuscript text omitted]

---

## Author Response (AR2)

**Dear Dr. Hoppema,**

We want to thank you for your help, and the three reviewers for their valuable comments about our manuscript. They have raised important points that helped us to improve clarity and became an article on a high level.
Below is a list of our correctness based on your comments.
All revised area is changed in the final manuscript with yellow highlighted.

**Topic Editor Decision: Publish subject to minor revisions (review by editor)** (07 Nov 2019)
by Mario Hoppema
Comments to the Author:
Dear Drs. Kim, Chapman and co-authors,

Your revisions are satisfactory and the manuscript is almost ready for acceptance. I went through it and below my final comments are listed.

Abstract L1 coastal sea (no capitals)
→ corrected
Abstr L2 nitrogen mass balance (not N here)
→ corrected
Abstr L5 no need for (AN-D); it is not used in the Abstract anymore
→ corrected
Abstr L6 I think "however" is not needed here.
→ deleted
Abstr L11 "more than 2" is not very precise and looks strange here. Can you please modify this?
→ we changed word "over 2 gC~"
L24 Please define SGD
→ L24, submarine groundwater discharge (SGD)
L67 I guess you mean high winds and high nitrate concentrations
→ L66-69, we changed the sentence more clearly.
L117 Please explain how data interpolation was done
→117-118, we explained more detail into the text
L124 Please change format: kg ha-1 year-1. Is this according to SI? ha is hectare, right? Even when this is the usual notation in this field of science, please also use SI units.
→ L125, corrected
L129 dito
→ L130, corrected
L145 ml/L is not a SI unit. Please (also) give the value as SI.
→ L146, corrected
L148-149 "which means that the two layers have different biological processes." I do not agree. The rates could just be different.
→ L149-150, corrected

L151-152 "While chlorophyll can be found below the pycnocline (DiMarco and Zimmerle, 2017)." This is not a complete sentence. Please correct.
→ L152-155, corrected, "While chlorophyll can be found below the pycnocline (DiMarco and

Zimmerle, 2017), the fact that it is typically associated with low oxygen concentrations suggests that the phytoplankton are either inactive or, more likely, producing at a very slow rate."

L208 "nitrate (NO2-) and nitrite (NO3-)" This is an error. It should be exactly the other way around.

→ L211, corrected

L277 Please use format 4 March 2005

→ L279-280, corrected

L295-296 The sentence was not clear. I suggest: We defined the brown zone as having the PPP rate of over 2 gC m-2 day-1 because …

→ L298-299, corrected

L328-329 … determined the integrated PP rates with 14C measurements during 2004 … (because twice in the sentence a word with "measure")

→ L331-332, corrected

L335 MR has not been defined before. Please consider not using that many abbreviations, because they reduce the readability.

→ Defined and corrected in L245

L508 suggest (instead of explained)

→ L507, corrected

L513-514 "in the near future both AN-D flux and riverine N flux need to be considered for managing nitrogen in coastal waters." This is double info and it appears in the previous paragraph and at other place in the manuscript.

→ deleted

L533-534 "We identified the brown zone close to the Keum River mouth and the green and blue zones further away from the coast of Korea." This is trivial and does not contain much information. If you would like to state something here, please give more details.

→ deleted

As to the references:

L555 No journal name

→ L557-563, corrected

L580 No journal name

→ L585-591, corrected

L685,688 Any more info on these reports? Report number, Place of publication?

→ L812-819, corrected

L723 Journal must be: Journal of the Atmospheric Sciences

→ L731-732, corrected

L731 upper case for 222

→ L739, corrected

L804 Any more info for this report?

→ L821-823, corrected

L855 Please change format of reference

→ L872, corrected

L893 Estuarine (typo)

→ L910, corrected

L929 Any more info for this report? Where available?

→ L945-947, corrected

Figure 1 caption. Please use Gulf of Mexico full, not GOM. a) Please add some geographical names in the figure for orientation. What does the drawn line represent? What about abbreviations and percentages? Where exactly is the sampling area? Is it the whole region? Please be more specific.

b) Please explain the colors

→ L962-969, corrected, "Study sites and sampling areas in the Gulf of Mexico and Korea. (a) shows the sampling area within the northern Gulf of Mexico. Flow in the Mississippi/ Atchafalaya River System is split 30% to the Atchafalaya River, 70% to the Mississippi River. The box is the sampling area. (b) shows station positions from March 2005. Note that MCH project data are widely distributed across the region. Red, grey, and blue stations correspond to sub-regions A (near the Mississippi River), B (between the Mississippi and Atchafalaya), and C (near the Atchafalaya) respectively. (c) shows the sampling area off the west coast of Korea. (d) shows all of the station positions."

L1020, 1022 delete "Figure"

→ Deleted

L1025 describes , not described

Figure 2 Please define RMEPs

→ L973, corrected

Figure 9 Please do not use abbreviations here

→ L1011-1012, corrected

Table 1 Please use data format like 5 April 2004

Table 4 Please use data format like 5 April 2004

L1084, 1088 There should be a minus sign in the ratio

→ Table 1, L1032-1035, 1039-1040, corrected

Thank you and best wishes

Mario Hoppema

[revised manuscript text omitted]